


# Towards the construction of regional marine radiocarbon calibration curves: an unsupervised machine learning approach

Ana C. Marza[1], Laurie Menviel[2], Luke C. Skinner[1]

[1]Department of Earth Sciences, University of Cambridge, Cambridge, CB2 3EQ, United Kingdom
[2]University of New South Wales, Sydney, AUS, Australia

*Correspondence to*: Luke C. Skinner (lcs32@cam.ac.uk)

**Abstract.** Radiocarbon may serve as a powerful dating tool in palaeoceanography, but its accuracy is severely limited by the need to calibrate radiocarbon dates to calendar ages. A key problem is that marine radiocarbon dates must be corrected for past

offsets from either the contemporary atmosphere (i.e. 'reservoir age' offsets) or a modelled estimate of the global average surface ocean (i.e. delta-R offsets). This presents a challenge because the spatial distribution of reservoir ages and delta-R offsets can vary significantly, particularly over periods of major marine hydrographic and/or carbon cycle change such as the last deglaciation. Modern reservoir age/delta-R estimates therefore have limited applicability. The construction of regional marine calibration curves could provide a solution to this challenge. However, the spatial reach of such calibrations, and their

robustness subject to temporal changes in climate and ocean circulation would need to be tested. Here, we use unsupervised machine learning techniques to define distinct regions of the surface ocean that exhibit coherent behaviour in terms of their radiocarbon age offsets from the contemporary atmosphere (R-ages). We investigate the performance of multiple algorithms (K-Means, K-Medoids, hierarchical clustering) applied to outputs from 2 different numerical models, spanning a range of climate states and timescales of adjustment. Comparisons between the cluster assignments across model runs confirm some

robust regional patterns that likely stem from constraints imposed by large-scale ocean and atmospheric physics (i.e. locations of deep mixing, gyres, fronts, divergence etc.). At the coarsest scale, regions of coherent R-age variability correspond to the major ocean basins (Arctic, Atlantic, Southern, Indo-Pacific). By further dividing basin-scale shape-based clusters into amplitude-based subclusters, we recover regional associations that cohere with known modern oceanographic processes, such as increased high latitude R-ages, or the propagation of R-age anomalies from regions of deep mixing in the Southern Ocean

to upwelling sites in the Eastern Equatorial Pacific. We show that the medoids (i.e. the most representative locations) for these regional sub-clusters provide significantly better approximations of simulated local R-age variability than constant offsets from the global surface average. This is found to hold when cluster assignments obtained from one model are applied to simulated R-age time series from another. The proposed clusters are also found to be broadly consistent with existing reservoir age reconstructions that span the last ~30 ka. We therefore propose that machine learning provides a promising approach to the

problem of defining regions for which marine radiocarbon calibration curves may eventually be generated.



## 1 Introduction

Radiocarbon was initially developed primarily as a dating tool (Libby, 1955); however, the conditions under which radiocarbon can be used to provide accurate calendar age dates turn out to be quite restricted. Due to past changes in the production rate of radiocarbon in the atmosphere, the total inventory of radiocarbon has changed over time. Furthermore, changes in the global carbon cycle have also caused changes in the distribution of radiocarbon amongst the various Earth system carbon reservoirs. Both of these factors have caused the radiocarbon concentrations of the various Earth system carbon reservoirs to change over time. The conversion of a radiocarbon measurement into a calendar age estimate requires knowledge of both the radiocarbon decay rate, or half-life, and the initial radiocarbon concentration of the fossil entity. Therefore, radiocarbon ages must be 'calibrated' to calendar ages using a reservoir-specific calibration curve that lists calendar ages and corresponding radiocarbon ages, derived from that reservoir's history of radiocarbon concentration change. Currently, the atmosphere is the only Earth system reservoir for which we possess a robust observation-based calibration curve (Reimer et al., 2020; Hogg et al., 2020). Even in the case of the relatively well-mixed atmosphere, subtle differences in the evolving radiocarbon concentrations of the northern and southern hemisphere atmosphere require the use of hemisphere-specific calibration curves (Reimer et al., 2020; Hogg et al., 2020).

The calibration of marine radiocarbon dates presents further challenges (Skinner and Bard, 2022). On the one hand, this is because the processes responsible for the exchange of radiocarbon between the ocean and atmosphere (where radiocarbon is produced), and the processes responsible for the redistribution of radiocarbon throughout the ocean, are relatively slow. This results in spatially heterogeneous patterns of radiocarbon concentration throughout the ocean, including in the 'surface' ocean (upper few 100m). **Figure 1a** shows the distribution of radiocarbon in the pre-industrial surface ocean, expressed in terms of radiocarbon age offsets between the ocean at a given location and the mean atmosphere (i.e. 'reservoir age' or R-age offsets). The patterns of radiocarbon concentration that are visible in **Figure 1a** are directly related to the oceanographic phenomena that control the ocean-atmosphere exchange of radiocarbon and its transport through the ocean (Key et al., 2004; Koeve et al., 2015).

From a calibration perspective, the problem that arises from this spatial heterogeneity is that the radiocarbon concentration at a given location cannot necessarily be estimated from e.g. the *mean* surface ocean radiocarbon concentration (or the mean offset from the atmosphere, R-age). One way to address this problem has been to apply a constant location-specific correction, referred to as a 'delta-R' correction (i.e. $dR(j)$, for location j) (Reimer and Reimer, 2001; Stuiver et al., 1986). Such delta-R corrections represent the difference between the local R-age a location $j$ and time $t$ (i.e. $R_{age}(t,j)$) and the mean surface ocean R-age at time $t$ (i.e. $\overline{R_{age}(t)}$):

$$R_{age}(t,j) = \overline{R_{age}(t)} + dR(j)$$





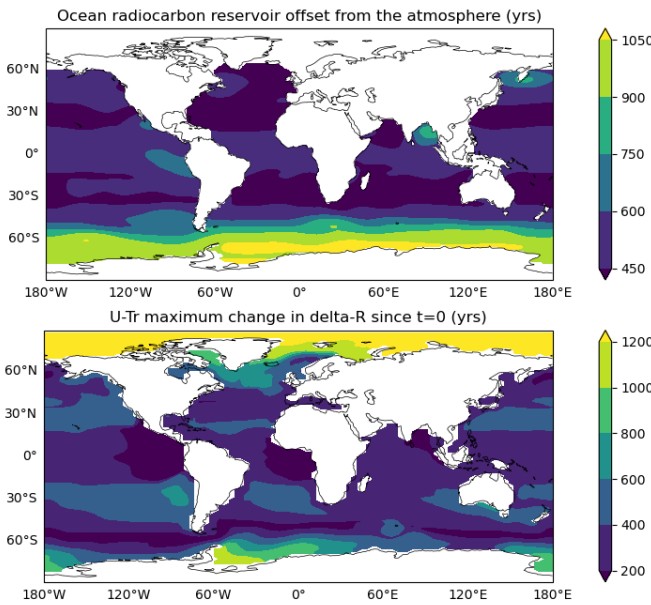

**Figure 1.** *Radiocarbon 'reservoir age' (R-age) offsets and their potential variability. Plot a, modern 'background' (bomb-corrected) R-ages averaged over the upper 300m, based on the GLODAP dataset (Key et al., 2004). Plot b, example of maximum changes in delta-R values (i.e. deviations between local R-ages and the global mean R-age) associated with ocean circulation changes simulated by the UVic Earth system model of intermediate complexity (Menviel et al., 2015).*

A marine radiocarbon date from location $j$ and time $t$ ($^{14}C_o(t,j)$) might therefore be expressed as:

$$^{14}C_o(t,j) = {}^{14}C_{atm}(t) + R_{age}(t,j) = {}^{14}C_{atm}(t) + \overline{R_{age}(t)} + dR(j)$$

Or equivalently, in terms of the mean surface ocean radiocarbon age ($\overline{^{14}C_{sfo}(t)}$):

$$^{14}C_o(t,j) = \overline{^{14}C_{sfo}(t)} + dR(j)$$

Therefore, assuming an invariant spatial distribution of R-ages (hence an invariant ocean state), one approach has been to apply a constant location-specific delta-R correction (usually based on modern pre-bomb estimates) to measured radiocarbon dates, and calibrate these corrected dates using a mean surface ocean calibration curve, such as *Marine20* (Heaton et al., 2020). The latter has been derived from the atmospheric calibration curve, by modelling the ocean's response to evolving atmospheric



radiocarbon concentrations (Heaton et al., 2020). One major drawback of this approach is that it requires assumptions (or faith in forward modelled outcomes) regarding past changes in key environmental parameters (such as sea ice distribution/seasonality, ocean circulation, global carbon cycling, etc.) that are actually the focus of intense debate and ongoing research efforts.

While the estimation of an appropriate mean surface ocean radiocarbon history (e.g. Marine20) presents significant challenges in itself, a further difficulty arises from the fact that local deviations from the global mean R-age (i.e. delta-R values) need not remain constant. Thus, a more accurate description of surface R-ages is:

$$R_{age}(t, j) = \overline{R_{age}(t)} + dR(t, j)$$

This additional challenge was already identified when the use of delta-R corrections was first proposed (Stuiver et al., 1986), at which time it was emphasized that their use was likely only justified over the last ~9000 years when the global carbon cycle and ocean state were thought to have remained 'more or less' constant. Indeed, it is now apparent that local R-age values have changed by order 100% over the last ~20,000 years at some surface ocean locations, and more importantly that such changes have followed different trajectories depending on their location (Skinner et al., 2019). Furthermore, as illustrated in **Figure 1b**, Earth system modelling results indicate that significant and spatially heterogeneous changes in delta-R values are expected to have occurred as a result of past ocean circulation perturbations (Menviel et al., 2015).

In order to circumvent these non-trivial and persistent challenges for marine radiocarbon dating, one approach would be to directly estimate past surface ocean radiocarbon variability, and thus reconstruct marine calibration curves, for key locations in the surface ocean (Skinner et al., 2019). This would circumvent the need to make assumptions regarding past ocean states and therefore substantially 'liberate' radiocarbon as a carbon cycle tracer as well as a dating tool. Already, the identification of regionally coherent patterns of R-age variability across the last deglaciation suggests that the construction of regional calibrations could be successful (Skinner et al., 2019). However, several questions arise in this context: how would 'regions' of coherent radiocarbon variability be defined; how robust and stable would such regions be, subject to major climate/ocean circulation change; what gain in calibration accuracy (if any) would be possible through their use? This study aims to address these questions. More specifically, we investigate the potential for regional marine radiocarbon calibrations using unsupervised machine learning techniques to define distinct regions of coherent R-age variability in the surface ocean. We apply our analysis to a suite of numerical model outputs representing a range of different climate- and ocean circulation states and compare our model-based results with currently available marine R-age reconstructions. Our results suggest that the investigated methods are promising as a way forward for the development and application of regional marine radiocarbon calibration curves.



## 2. Methods

We apply unsupervised machine learning (ML) techniques to outputs from 2 different Earth System models (section 2.1, Table 1). Our aim is to identify surface ocean locations that exhibit similar R-age variability, in terms of both their signal (or 'shape') and their amplitude. We make use of three different techniques for clustering (K-means, K-medoids and hierarchical clustering), in addition to a suite of methods for assessing their robustness. These are described below in section 2.2.

*Table 1: Summary of model outputs used.*

| Model | Description | Runs used | Time interval | Grid resolution (lon x lat) |
|---|---|---|---|---|
| CM2Mc.v2 | Coupled ocean-atmosphere-ice-biogeochemistry general circulation model, forced by CO$_2$, orbital configuration, ice sheet size (Galbraith and De Lavergne, 2019). | • **Glacial**-like equilibrium state <br> • **Interglacial**/pre-industrial-like equilibrium state. | 1 year (12 time steps, 1 month each) | 120 x 80 |
| UVic Earth System Model v2.9 (Menviel et al., 2015). | Transient simulations of MIS3 millennial-scale climate variability, with freshwater forcing periodically applied to the North Atlantic region to simulate Dansgaard-Oeschger events. | • **U-Tr** (UVic transient) <br> • **U-TrS** (additional transient experiment imposing a salt flux in the Southern Ocean and the Eastern Equatorial Pacific to correct for model limitations in representing the hydrological cycle). | 154 time steps (100 years each) spanning the interval from ~50 ka to ~34 ka. | 100 x 100 |

### 2.1 Model data

In order to explore regional associations in R-age variability, we make use of outputs from two different numerical models: CM2MC and UVic (**Table 1**). For the CM2Mc simulations, we use annual cycles drawn from an equilibrium interglacial-like state and from an equilibrium glacial-like state (Galbraith and De Lavergne, 2019). The UVic outputs consist of annual averages (at 100 year resolution) from two sets of transient simulations performed under 'mid-glacial' boundary conditions equivalent to Marine Isotope Stage (MIS) 3 (Menviel et al., 2015). One of these simulations (U-Tr) involved variable buoyancy forcing applied to the North Atlantic, resulting in changes in the strength of the Atlantic Meridional Overturning Circulation (AMOC). A second set of UVic simulations (U-TrS) involved in addition variable buoyancy forcing applied to the Indian and Pacific sectors of the Southern Ocean, as well as the Eastern Equatorial Pacific (EEP), resulting in additional changes in the strength of deep convection near Antarctica and in the North Pacific (Menviel et al., 2015). The Uvic simulations consist of two consecutive runs, hence four ocean circulation perturbations, for each of the UTr and UTrS simulations. The start of the second run was initialised far from equilibrium for radiocarbon, and therefore includes a global spike in marine radiocarbon activity. This has been retained in our analyses to permit an assessment of our ability to identify regional clusters both with and without the occurrence of a large background global anomaly, such as would be produced by the a large geomagnetic excursion for example (Heaton et al., 2021).



The dissolved inorganic carbon (DIC) and the dissolved inorganic radiocarbon (DI$^{14}$C) to compute surface ocean $\Delta^{14}$C:

$$\Delta^{14}C = \left(\frac{DI^{14}C}{DIC} - 1\right) \times 1000$$

R-age offsets are then calculated, taking into account the atmospheric $\Delta^{14}C_{atm}$:


$$R_{age} = -8267 \times \ln\left[\left(\frac{\Delta^{14}C_{ocean}}{1000} + 1\right) \middle/ \left(\frac{\Delta^{14}C_{atm}}{1000} + 1\right)\right]$$

Here we use the 'true' mean lifetime of radiocarbon (8267 years) based on the 'Cambridge half-life' of 5730 years (Godwin, 1962). However, we note that if a direct quantitative comparison with measurements was to be performed, then the

conventional 'Libby half-life' of 5648 years should be used instead. The atmospheric $\Delta^{14}$C in the CM2Mc runs is held constant at 0 per mil, and in the UVic runs at 393 per mil.

## 2.2 Unsupervised machine learning

### 2.2.1 K-means

The K-means algorithm (see Ahmed et al. (2020) for a recent review) divides data into a number K of clusters, defining the partitions such that each data point is as close as possible to the mean of its assigned cluster (**Figure 2**). For our purposes, the R-age time series of each ocean location on the model grid constitutes one data point for K-Means, letting us map out which grid points are assigned to which cluster. We feed no prior geographical information to the algorithm; therefore, regions delineated by the algorithm are based on the similarity of their R-age histories alone.


### 2.2.2 K-Medoids

The K-Medoids algorithm is similar to K-Means, but, instead of converging on abstract cluster centroids, it identifies cluster medoids, i.e. actual data points that best represent each cluster. K-Medoids is more robust to noise and outliers in the data (Arora et al., 2016); furthermore, working with concrete data points as cluster centres, we can pinpoint their locations on the

model grid. In the implementation we used, K-Medoids was slower than K-Means, so although most of the results section focuses on K-Medoids, much of the exploratory analysis involved K-Means.

### 2.2.3 Selecting K for K-means and K-medoids

The main drawback of both K-Means and K-Medoids is that they require a priori knowledge of the number of clusters, K, to

divide data into. We employ 4 methods (Table 2) to evaluate clustering performance as a function of the number of clusters over the range $2 \leq K \leq 10$. The Caliński-Harabasz (CH) index and the silhouette score penalize different aspects of cluster




shape: CH generally increases with K, because it favours splitting the data into many small clusters to minimize intra-cluster distances; the silhouette score generally decreases with K, because it favours a few large, well-separated clusters.

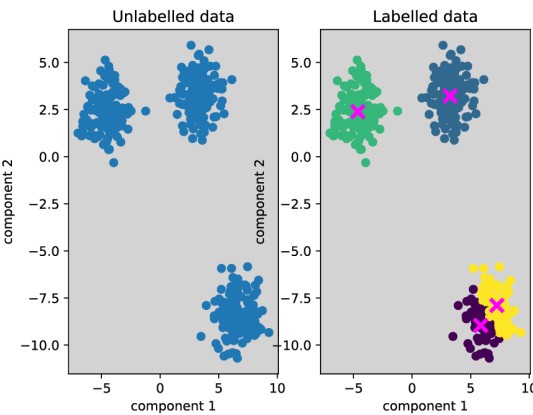

**Figure 2.** *Illustration of the basic operation of the K-means clustering algorithm, on synthetic data. Each data point is defined here by two scalar values corresponding to two component axes. The unlabelled data on the left are assigned to four colour-coded clusters on the right. The cluster centroids are marked by crosses.*

Table 2: Methods to pinpoint the optimal number of clusters.

| Method name | Description | Reference |
|---|---|---|
| The elbow/knee method | This heuristic technique simply considers as optimal the number of clusters for which the sum of squared errors (distances), SSE, between cluster members and their respective cluster centres is no longer significantly reduced by increasing K (i.e. we have 'diminishing returns' in error reduction as K increases). | Satopaa et al. (2011) |
| Caliński-Harabasz index | Captures intra-cluster cohesion and inter-cluster separation. | Caliński and Harabasz (1974) |
| Davies-Bouldin index | | Davies and Bouldin (1979) |
| Silhouette score | | Rousseeuw (1987) |

### 2.2.4 Time series normalisation and subclusters

We run K-Means and K-Medoids clustering on both unnormalised ('raw') and normalised time series. In the first case, clustering captures differences in both the amplitudes and shapes of the time series. In the latter case, only shape information is considered.

Because clustering on normalised time series disregards amplitude information, we perform a first round of clustering using normalised data to identify time-series that share the same patterns of variability, regardless of amplitude. A second round of





clustering, using un-normalised data, is then performed on each of these 'shape based' clusters. This allows us to identify amplitude-based 'subclusters' within the shape-based clusters. The 'optimal' number of subclusters within each cluster is
chosen at runtime using the elbow method (see section 2.2, Table 2).

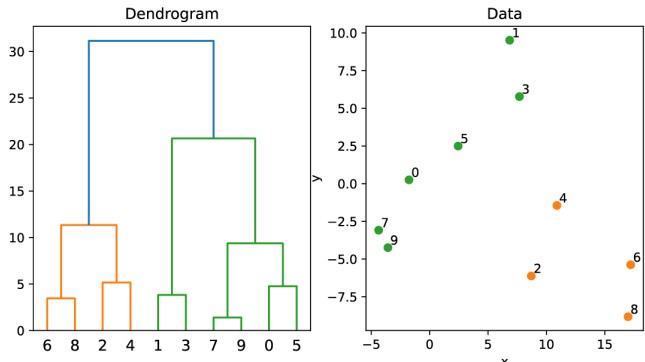

***Figure 3****. Illustration of the basic operation of the hierarchical clustering algorithm, on synthetic data defined along two component axes. Note how points which plot closer together in the right-hand plot are recovered as more closely related in the dendrogram on the left.*


### 2.2.5 Hierarchical clustering

Hierarchical clustering tells us how closely related data points are to each other, like phylogenetic trees. This requires the definition of an appropriate relatedness metric (or conversely, a distance metric). Using the Pearson correlation coefficient between the R-age histories of sites A and B in the ocean ($\rho_{AB}$), we may define a distance metric, $d = \sqrt{1 - \rho_{AB}}$ . This
distance metric provides a measure of the similarity of two time-series: d = 0 when $\rho_{AB}$ = 1 (the distance between perfectly correlated time series is zero); d = 1 when $\rho_{AB}$ = 0 (uncorrelated time-series are farther apart). The square root ensures that the triangle inequality is obeyed (Solo, 2019), avoiding misleading results (for example, when we fail to enforce the triangle inequality, we may obtain that time series A is similar to B, and B is similar to C, but A and C are dissimilar).

Given a matrix of distances between data points, there exist numerous methods to form a hierarchy/dendrogram (**Figure 3**). We opt for the Ward method, which outperforms other common linkage methods where clusters overlap (Vijaya et al., 2019), as expected for our data. The resulting dendrogram illustrates cluster relatedness and has the advantage (over K-Means and K-Medoids) of not requiring prior information on the number of clusters. The dendrogram can be split ('flattened') into an arbitrary number of clusters, by cutting the tree at any height.


### 2.2.6 Cluster analysis

The numerical cluster labels generated by the clustering algorithms are assigned randomly, hampering comparison between applications. To bypass this limitation, we: 1) re-order cluster labels such that clusters with higher mean R-ages are assigned

smaller numbers; 2) define a re-labelling routine that takes in two sets of clustering results and attempts to permute the labels

to maximise geographic overlap in cluster assignment between the two maps. Cluster maps with different grid sizes are re-gridded using a nearest-neighbour algorithm, and invalid data points (e.g. re-gridded onto land) are dropped from the analysis. As discussed below in section 3, a 1-to-1 matching between the two sets of labels is not always possible.

Given a set of subclusters, we investigate to what extent the R-age time series belonging to each subcluster are described: 1)

by the subcluster medoid; 2) by the 'parent' shape-based cluster medoid; and 3) by the running mean R-age of the surface ocean. We take the difference between each of these 3 'benchmark' R-age time series and the subcluster member time series at each point in time, obtaining a distribution of R-age anomalies for each of the three cases. We compare the three anomaly distributions graphically as density plots, and numerically in terms of their means.

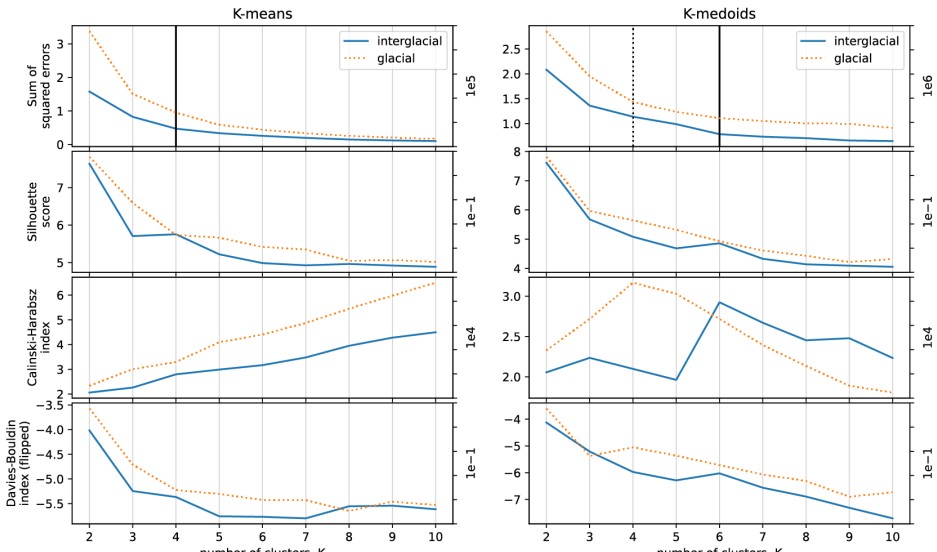

***Figure 4.*** *Clustering performance of K-means (left) and K-medoids (right) on un-normalised data, as a function of the number of clusters over the range 2 ≤ K ≤ 10, for the CM2Mc runs using unnormalized data (solid line for interglacial data; dotted line for glacial data). Vertical lines mark the elbow points. Note how, for the interglacial case, the elbow method and all three scores recover K=4 as a peak in clustering performance with K-means, K=6 with K-medoids.*

## 3. Results

### 3.1 Equilibrium annual cycle for interglacial- and glacial-like states

**Figure 4** summarises the K-means and K-medoids clustering performance for different numbers of clusters (K), between K=2 and K=10. The clustering is applied to one annual cycle of un-normalised (raw) data from an interglacial-like climate state





(solid lines in **Figure 4**) and a glacial-like climate state (dashed lines in **Figure 4**) simulated by the CM2Mc model (Galbraith
and De Lavergne, 2019). Note that the 'Davies-Bouldin' index has been multiplied by -1 to match the 'Caliński-Harabasz'
index and the silhouette score, such that lower values indicate diminishing explanatory power. For K-Means, the 4 methods
suggest K=4 as a particularly suitable number of clusters; it is pinpointed by the 'elbow method' (vertical dashed line) and
rises above the background trends in the silhouette score, the CH index (slightly) and the DB index. A similar result is obtained
for the glacial like climate state (**Figure 4**, left hand plots, dashed lines). There is less agreement among the methods using K-
Medoids for the interglacial-like climate state (**Figure 4**, right hand plots, solid lines), with an elbow at K=5, a small peak in
silhouette scores at K=6, strong peaks in the CH index at K=6 and K=8, and peaks in the DB index at K=5 and K=9. When
applied to raw data from the glacial-like climate state, K-medoids yields peaks in the CH and DB indices at K=4 and K=9
(**Figure 4** left hand plots, dashed lines). For both K-means and K-medoids, no clear optimum value for K emerges for K>2
when using normalised data (**Figure 4**), though it should be noted that the 'optimal' K values identified in this way are generally
rather qualified. This remains true when applied to un-normalised data from within each of the clusters identified using
normalised data (not shown).

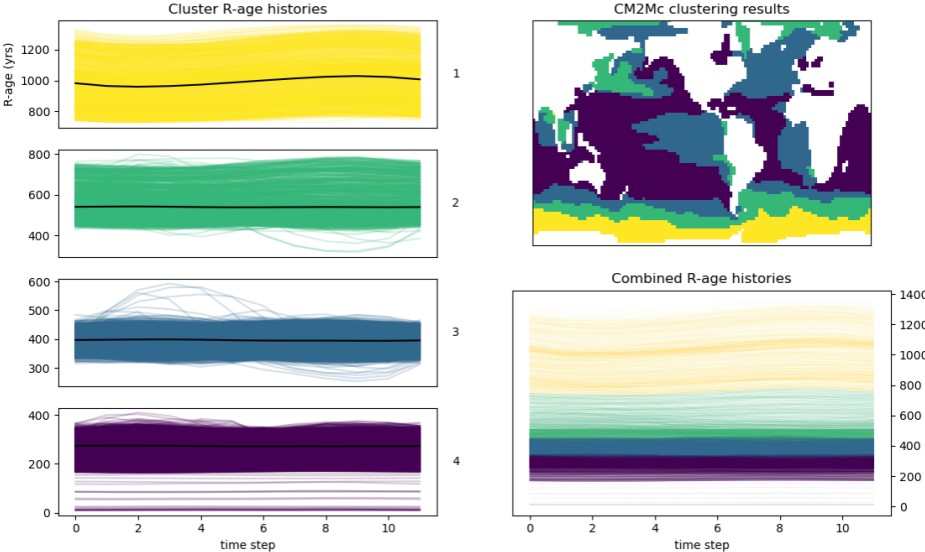

**Figure 5.** *Clustering results using K-medoids on un-normalised data from the CM2Mc interglacial run, using K=4. For*
*explanation see text.*


In this context, where we are interested in the eventual construction of regional radiocarbon calibration curves, an optimal
value of K will be the *minimum* value that provides a sufficient degree of explanatory power. Here, explanatory power reflects
the degree to which the cluster centroid/medoid is able to represent the R-age history at a given location with sufficient
accuracy and precision. Encouragingly, **Figure 4** suggests that, for two very different equilibrium climate states, the bulk of



R-age variability can be captured with a relatively small number of clusters, e.g. K=4, and with little sensitivity to clustering method (e.g. K-means or K-medoids).

**Figure 5** shows the clustering results for K-Medoids applied to un-normalised CM2Mc interglacial data, using K=4. The cluster map in **Figure 5a** illustrates the ocean regions corresponding to each cluster (landmasses in white); **Figure 5b** shows
the time-series associated with each cluster along with their respective centroids; and **Figure 5c** shows all the R-age time-series colour-coded by cluster membership. Cluster #1 forms a longitudinal band in the Southern Ocean, featuring the highest mean R-ages and the strongest annual variation in R-age. Clusters #2 and #3 cover lower-latitude band within the Southern Ocean, but also outcrop in the Eastern Equatorial Pacific (EEP), the Eastern Central Atlantic, the North Atlantic, the North Pacific and the North Indian Ocean. The rest of the ocean is assigned to Cluster #4, which dominates the tropical/subtropical
gyre regions and exhibits the lowest R-ages and the least variability over the annual cycle. Increasing the number of clusters to K=9 further subdivides the ocean (**Figure S1**), but with only a modest gain in explanatory power, producing the same broad patterns as for K=4.

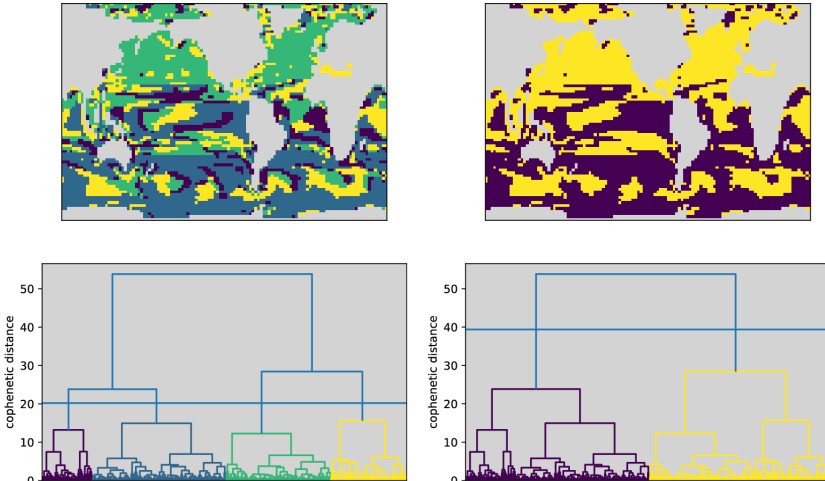

*Figure 6. Hierarchical clustering results for the CM2Mc interglacial run, based on Pearson distances between the R-age time*
*series at each location. Bottom panels show the same cluster dendrogram but sliced at two different heights (cophenetic distances) to produce four and two clusters on the left and right, respectively. Top panels show the resulting cluster geographies, with landmasses in grey.*

Clustering on normalised data, to extract clusters based on time series shape alone (without amplitude information), produces
highly geographically disconnected clusters for both K-Means and K-Medoids when K>2 (not shown). Nevertheless, a north-south divide is apparent in the clustering results for normalised data using K=4, reflecting an annual signal that is dominated by high-latitude seasonal convection and ocean-atmosphere gas-exchange (also apparent in the centroid time-series shown in



**Figure 5b**). The emergence of this north-south divide is further illustrated by hierarchical clustering on Pearson distances, which is also relatively insensitive to time-series amplitude and recovers similar regions to the shape-based clustering (**Figure**
**6**). The resulting dendrogram confirms that the north-south divide emerges at the highest branching level (**Figure 6b**).

Applying K-medoids to the glacial-like climate state simulated by CM2Mc, again using K=4, yields very similar (though not identical) results. This is shown in **Figure 7**, which compares the clusters obtained for the glacial- and interglacial states simulated using CM2Mc, demonstrating ~80% overlap. The similarity of the clusters obtained for the two simulations suggests
minor differences in the overall impact of the annual cycle on the distribution of R-ages between two contrasting climate states. However, it is notable that the regions of non-overlap tend to occur at the margins of the clusters, suggesting an inherent ambiguity at the junctures of cluster regions (hatched areas in **Figure 7**).

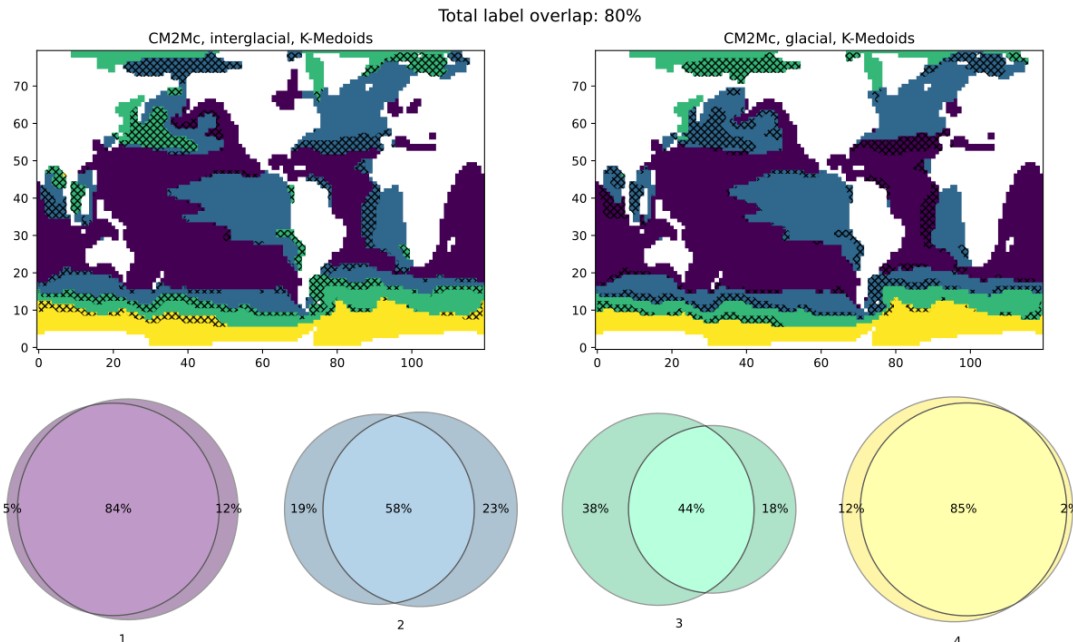

***Figure 7.*** *Comparison of clustering results between the two CM2Mc runs (left: interglacial; right: glacial), using K-medoids on un-normalised data. The colour-coded Venn diagrams quantify overlap between the two maps. For example, cluster #3 (green) is assigned substantially more area in the interglacial run on the left, while cluster #4 (yellow) covers roughly the same geographic region in both runs.*




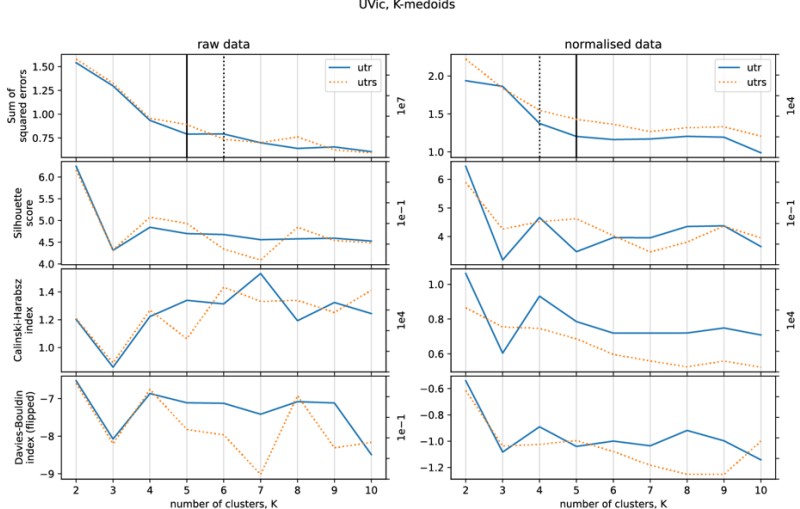

***Figure 8.*** *Clustering performance of K-medoids on un-normalised (left) and normalised (right) data, as a function of the number of clusters over the range 2 ≤ K ≤ 10, for the UVic runs (solid line – U-Tr data; dotted line – U-TrS data). Vertical lines mark the elbow points.*


### 3.3 Transient freshwater 'hosing' experiments

**Figure 8** illustrates K-Medoids clustering metrics for raw and normalised data (mean annual, in 100 year time steps) from the U-Tr and U-TrS simulations, which involved transient buoyancy forcing applied in the North Atlantic, as well as the Southern Ocean and EEP, against a 'mid-glacial' background climate state equivalent to Marine Isotope Stage (MIS) 3 (Menviel et al.,

2015) (see **Table 1**). Similar to the CM2Mc results, optimal values of K for the UVIC outputs appear to lie at 4, 8 or 9. For comparison with the CM2Mc clustering results on un-normalised data, we choose to cluster the U-Tr and U-TrS data using K=4.

The U-Tr clustering results using K-Medoids on normalised data (K=4) are shown in **Figure 9**. Broadly similar results are
obtained when using K-means instead (not shown). The regional clusters that emerge from the U-Tr data broadly correspond to the major ocean basins (**Figure 9a**): The Arctic Ocean (Cluster #2) shows strong R-age variability; the Atlantic Ocean (#3) has smaller R-age peaks; the R-age time series of the Southern Ocean (#1) are rather flat compared to the others, while the Indo-Pacific cluster (#4) features broader and less accentuated maxima (**Figure 9b**). Again, increasing the number of clusters to K=8 results in further splitting of the ocean basins (e.g. isolating the North Atlantic and the southern half of the Pacific), but
with broadly the same basin-wide divisions as for K=4 (**Figure S2**).





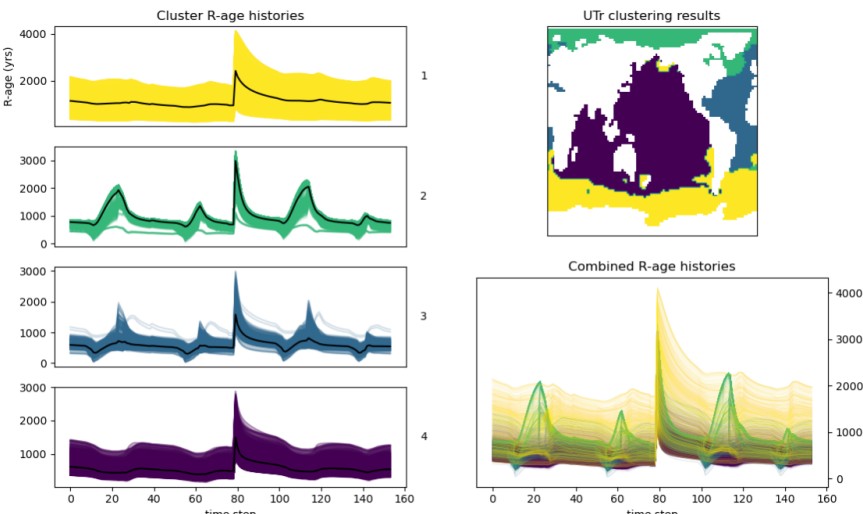

**Figure 9.** *Clustering results using K-Medoids on UVic (U-Tr run) normalised data. Layout same as for Figure 5. Note how clustering on normalised data groups together similarly shaped R-age histories, regardless of their amplitudes.*


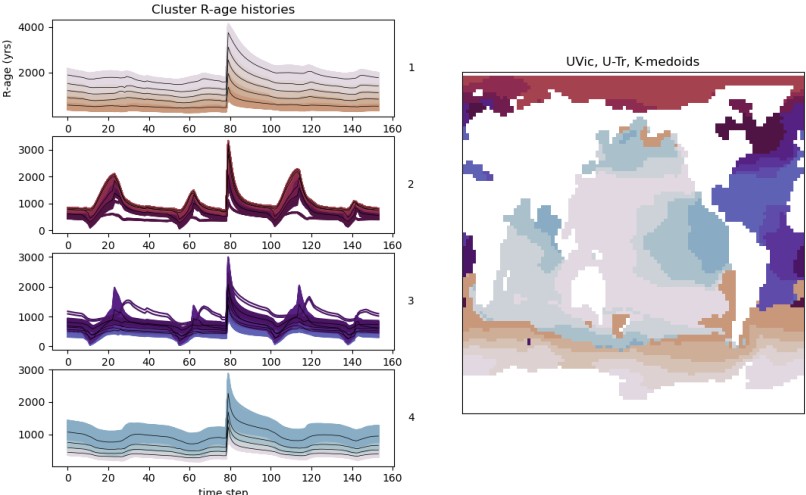

**Figure 10.** *Clustering results using K-Medoids on UVic (U-Tr run) data, in two stages, whereby the shape-based clusters from Figure 9 are further subdivided by re-applying K-Medoids to un-normalised data. Left plot: R-age time-series grouped by cluster 1-4, with sub-clusters identified by shading and sub-cluster medoids shown by heavy green lines. Right plot: Map*
*of R-age clusters (identified by colours) and sub-clusters (identified by shades of each colour). Note how the subclusters are distinguished by their mean R-ages.*





To explore the possibility of identifying sub-regions of similar amplitude variability, within each shape-based cluster, we perform a second round of clustering on un-normalised data from within each of the four shape-based clusters shown in **Figure 9**. The results are shown in **Figure 10**, demonstrating a decrease in the amplitude of R-age variability with decreasing latitude in the Southern Ocean, the Arctic Ocean, and the North Atlantic, despite distinct patterns of variability in each of these regions. Similarly, there is a decrease in R-age amplitude away from some ocean margins, e.g. in the EEP, the Eastern Central Atlantic, the North Pacific, and the northern Indian Ocean.

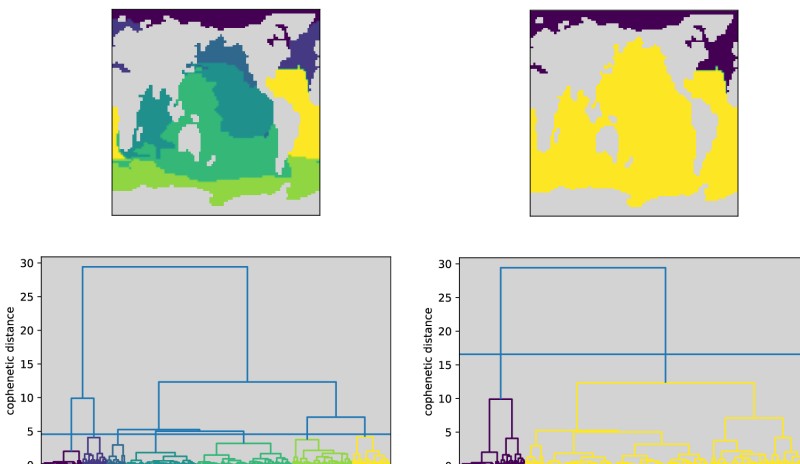

**Figure 11.** *Hierarchical clustering results for the UVic U-Tr run. Bottom panels show the same cluster dendrogram but sliced at two different heights (cophenetic distances) to produce seven and two clusters on the left and right, respectively. Top panels show the resulting cluster geographies, with landmasses in grey.*

Like with the CM2Mc data, hierarchical clustering of the U-Tr data based on Pearson correlations (**Figure 11a**) produces similar geographic patterns to K-Medoids clustering on normalised data. The Arctic Ocean is most closely related to the North Atlantic on the dendrogram. Meanwhile, the Central Atlantic is more related to the Southern Ocean than to the North Atlantic, marking a north-south divide high up on the dendrogram highlighted in **Figure 11b**. The North Pacific stands out from the rest of the Pacific, but overall the Pacific appears to be more closely related to the central Atlantic and Southern Ocean than to the Arctic and North Atlantic.

A comparison of results obtained for normalised data from U-Tr (North Atlantic 'hosing') and U-TrS (North Atlantic 'hosing' with Southern Ocean buoyancy forcing) indicates cluster overlap ~95% (**Figure 12**). The main differences occur along the northern margin of the Southern Ocean (hatched area in **Figure 12**). The difference between the two UVIC simulations is smaller than the difference between the two CM2Mc simulations. However, a comparison of clusters obtained using K-medoids on un-normalised data from the U-Tr simulation (transient annual average) and the glacial CM2Mc simulation (equilibrium annual cycle) yields an overlap of ~70% (**Figure S3**). Here the main difference arises from the identification of





a region of distinct annual variability in the sub-polar Southern Ocean in the glacial CM2Mc simulation, which is not present in the transient U-Tr simulation. Nevertheless, the high degree of overlap between two different models and two very different timescales of variability is notable.

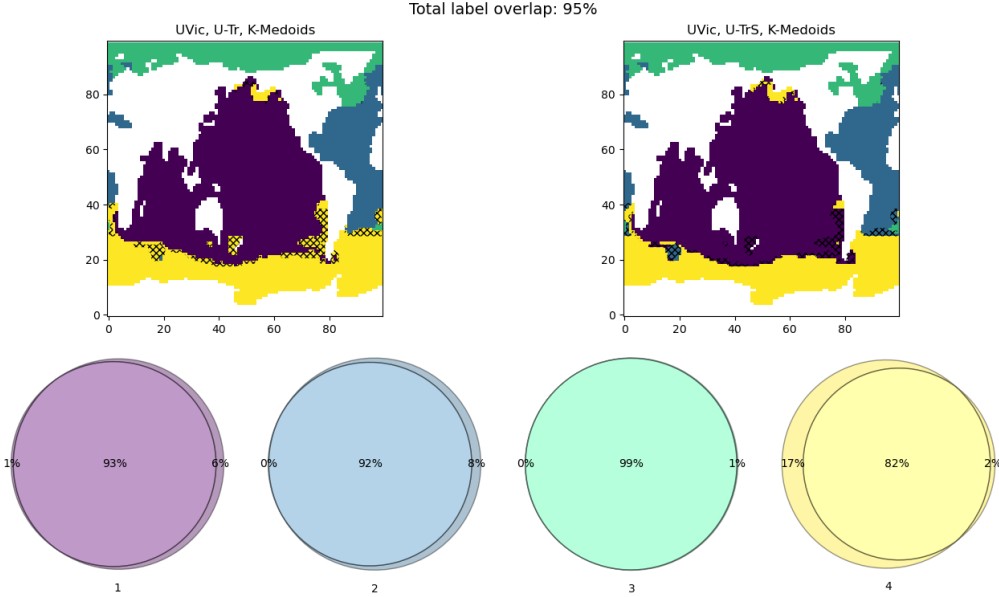


**Figure 12.** *Comparison of clustering results between the two UVic runs (left: U-Tr; right: U-TrS), using K-medoids on normalised data. The colour-coded Venn diagrams quantify overlap between the two maps. Note the excellent agreement between the two resulting cluster geographies.*

## 4. Discussion

The broad similarity of the clustering results obtained across the suite of equilibrium- and transient simulations (**Figures 7, 12 and S3**) suggests that the regional R-age associations arise to a large extent from fundamental aspects of global ocean/climate dynamics. For example, the hemispheric partitioning in normalised annual cycle data from both glacial and interglacial climate states clearly reflects the dominant influence of the annual cycle. Similarly, the regional clusters obtained in un-normalised

annual cycle data (**Figure 5**), and especially in transient annual average data (**Figure 9**), appear to reflect provinces of distinct hydrographic variability that are defined by fundamental oceanographic characteristics such as the presence/absence/variability of e.g. sea-ice, deep mixing, or upwelling. Such features are typically geographically 'locked', despite being time-variant in their intensity/expression. It was an appreciation of such regional specificity in the mechanisms that control R-age variability that provided an initial justification for using constant corrections to the global mean R-age (i.e. delta-R values) as an

approximation of past local R-age variability (Stuiver et al., 1986). R-ages in regions with extensive sea ice cover, deep mixing or upwelling will always be offset to higher values as compared to the global mean for example. In the UTr and UTrS





simulations, the Arctic and Antarctic R-age clusters broadly coincide with regions of similarly coherent sea-ice variability (not shown), though the overlap is not perfect suggesting that a more complex array of processes controls the regionalism in R-age variability. The clustering results for the CM2M2 simulations suggest that regional seasonality (in sea ice, mixed layer depth etc.) may play a contributing role. Here, our primary interest is in the viability of the clustering approach when applied to realistic (i.e. modelled) R-age variability, rather than the precise reasons for the variability that emerges in the model simulations that are selected. Nevertheless, a detailed analysis of the controls on regional R-age variability in model simulations that aim specifically to reproduce R-age reconstructions represents an important target for future work. Here too, clustering may yield useful insights.

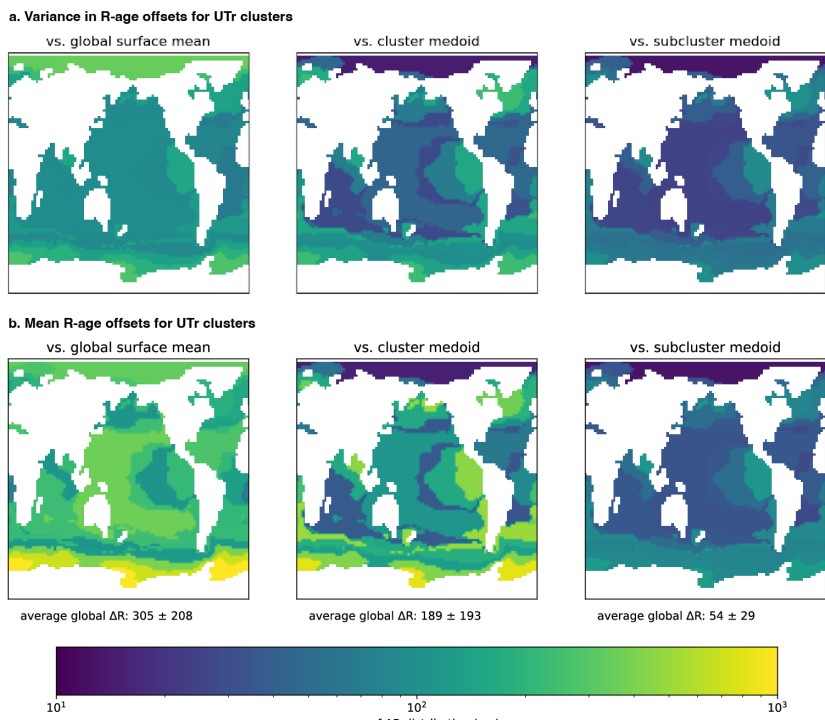

***Figure 13.*** *The variance (a.) and mean (b.) of the distribution of R-age offsets in the UVic U-Tr data, when computed relative to three different references: the running mean of the global surface ocean, analogous to e.g. Marine20 (column 1); the medoid of the shape-based cluster to which the subcluster belongs (column 2); and the subcluster's own medoid (column 3). The decreasing magnitudes left-to-right suggest an increase in the accuracy with which we describe the reference describes R-age histories in each subcluster, when moving from a mean ocean reference to the sub-cluster's medoid. This suggests that regional calibrations, e.g. for cluster or sub-cluster regions, could provide more accurate calibrations than constant delta-R values applied to a global mean estimate such as Marine20.*





In addition to stable 'regionality' of R-age behaviour, the use of delta-R values over long spans of time further requires that regional R-ages follow the same temporal trends as the global mean. As noted when the delta-R approach to marine radiocarbon calibration was formulated (Stuiver et al., 1986), and as illustrated by the U-Tr and U-TrS simulations (**Figures 9 and 10**), such global coherence cannot necessarily be expected when the ocean-climate system undergoes significant change (see also **Figure 1b**). Changes in sea ice cover, mixed layer depth, upwelling strength, etc., will change the difference between a local

R-age and the global mean (i.e. delta-R). Therefore, a key question for radiocarbon calibration is whether regional clusters (and sub-clusters), as identified here, can provide more accurate estimates of past R-age variability at a given location than is provided by e.g. the global mean (± a constant correction, delta-R).

Here, this question can be answered for the model outputs by comparing the magnitude, and in particular the variance, of

offsets between local R-ages and: 1) the global mean R-age; 2) the associated (shape-based) cluster centroid; and 3) the (amplitude-based) sub-cluster centroid. The variance in these offsets is important because it reflects the degree to which a constant correction, e.g. applied to a global average calibration curve (a delta-R value), would be wrong on average. **Figure 13** illustrates the spatial distributions for each of these offsets and their variance, based on the U-Tr outputs.  In **Figure 13**, the largest R-age offsets and the greatest variance in R-age offsets occur when referencing to the global mean surface R-age (305

± 208 $^{14}$Cyrs). The smallest magnitude and variance occur when referencing to the centroid of the relevant amplitude-based sub-cluster (54 ± 29 $^{14}$Cyrs), and an intermediate gain in accuracy is achieved when referencing to the centroid of the wider shape-based cluster (189 ± 93 $^{14}$Cyrs). Encouragingly, a similar stepwise improvement in accuracy is found when shape-based clusters or amplitude-based sub-clusters from the U-Tr simulation are used to assess R-age offsets in the glacial CM2Mc simulation (e.g. average offsets of 259 ± 227 $^{14}$Cyrs for the global mean *versus* 79 ± 36 $^{14}$Cyrs for sub-cluster centroids; **Figure**

**S4**).

The above discussion would suggest that radiocarbon calibrations performed using a regional calibration curve, particularly one derived at an appropriate sub-cluster centroid location, could be more accurate than calibrations performed using a global mean calibration curve in conjunction with a constant delta-R value. By way of illustration, a R-age (or delta-R) uncertainty

of 200 (versus 30) $^{14}$C yrs would result in a calibrated age uncertainty of ~540 (versus ~170), when calibrating a radiocarbon date of 20,000 ± 150 $^{14}$C yrs.  Similarly, a delta-R error/bias of up to ~1000 $^{14}$C years, as is observed at high southern latitudes in the UVic simulations (**Figure 1b**), would result in similar calibrated age error/bias.  Notably, the above analysis likely underestimates the uncertainty associated with using a global mean calibration curve and constant delta-R value in practice, since it assumes perfect knowledge of the evolving global mean R-age (i.e. a global mean radiocarbon calibration).  As noted

above, our best estimate of the mean ocean R-age history is currently based on modelling and therefore assumptions regarding past ocean circulation and climate change (Heaton et al., 2020).



While our analysis provides an illustration of the viability and the utility of defining regions for which local calibrations might be constructed, it only does so theoretically, using model outputs. However, the fact that regional clusters can successfully be

identified in model outputs is in itself consistent with the observation of distinct regional patterns in R-age reconstructions, e.g. from the Northeast Atlantic, Iberian Margin, South Atlantic, and Southern Ocean (Skinner et al., 2019). The further observation of similar but lower amplitude R-age trends on the Iberian Margin *versus* the northern Northeast Atlantic, and in the South Atlantic *versus* the Southern Ocean, also resonates with the sub-cluster results obtained from the U-Tr and U-TrS outputs, as does the observation of an apparent link between the high Southern latitudes and the eastern equatorial Pacific (De

La Fuente et al., 2015) (**Figure 12**).

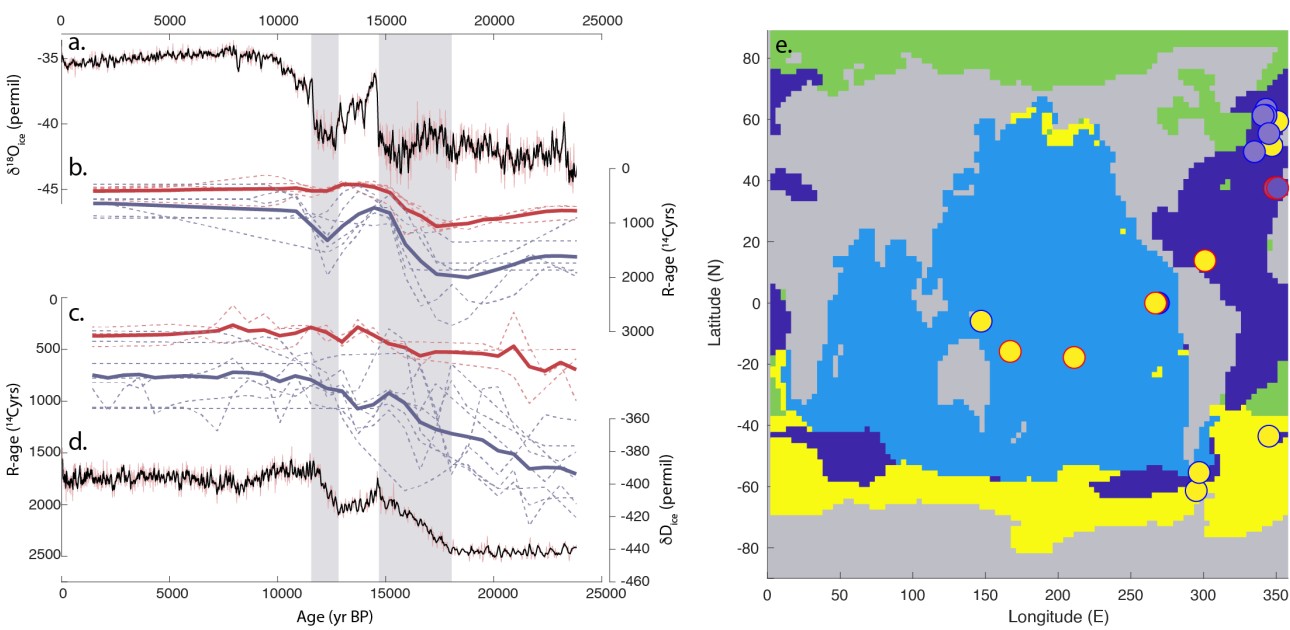

***Figure 14.*** *Tentative hierarchical clustering of R-age reconstructions spanning the last deglaciation (Skinner et al., 2023). Plot a: Greenland ice-core temperature proxy, NGRIP $\delta^{18}O_{ice}$ (heavy black line is 3-point running mean). Plot b: amplitude-*

*based sub-clusters (K = 2, red and blue lines) belonging to shape-based Cluster 1; heavy lines indicate the sub-cluster averages and dashed lines indicate the clustered time-series data. Plot c, as for plot b, but for sub-clusters within the shape-based Cluster 2. Plot d: Antarctic ice-core temperature proxy, EDC $\delta D_{ice}$ (heavy black line is 3-point running mean). Plot e: R-age time-series locations and cluster membership superposed on a map of shape-based hierarchical clusters for the UTrS simulation for comparison; purple filled circles indicate cluster 1 (plot b), yellow filled circles indicate cluster 2 (plot c), and*

*red/blue outlines indicate low/high amplitude sub-cluster membership. Shape-based cluster 1 (purple circles) overwhelmingly comprises time-series from the northeast Atlantic, with lower- and higher- latitude data included in separate amplitude-based subclusters.*





The regional patterns picked out by clustering of model outputs are further confirmed by a tentative hierarchical clustering
performed on 23 available shallow sub-surface R-age reconstructions from <500m water depth, or <1000m in the Southern
Ocean. For this analysis, time-series were selected that include at least 6 data points from the 5 – 21 ka BP interval, which
were binned and interpolated onto 500 yr intervals. Using normalised data and K = 2 (given the lack of data from e.g. the
Arctic, Indian, or North Pacific), hierarchical clustering identifies a dominant signal in the North Atlantic (**Figure 14b, e**) that
is distinct from the primary signal observed in the Southern Ocean and Pacific (**Figure 14c, e**). Amplitude-based sub-clusters
from within these regional shape-based clusters (again with K = 2), further isolate a higher amplitude high-latitude Northeast
Atlantic signal (**Figure 14b**, blue line; and **Figure 14e**, purple circles with blue outline) from a lower amplitude Iberian Margin
signal (**Figure 14b**, red line; and **Figure 14e**, purple circles with red outline). This is in keeping with the model clusters (e.g.
**Figure 10**). Higher amplitude Southern Ocean records are also grouped together (**Figure 14c**, blue line; and **Figure 14e**,
yellow circles with blue outline), though the method struggles to separate these from a few low-latitude and Northeast Atlantic
records that are included in the same cluster.  As shown in **Figure 14**, the two main signals identified by the hierarchical
clustering appear to track Greenland and Antarctic temperature variability, as highlighted by Skinner et al. (2019).  Although
the similarities between data- and model-based regional clustering are encouraging, it should be noted that the cluster results
for observations (**Figure 14**) are not especially robust, and are highly sensitive to parameter selection (e.g. K, minimum time-
series length), as well as the selection of data for inclusion.  A far larger number of better-resolved surface reservoir age data,
spanning a greater geographical range, will be needed to improve upon the highly tentative data-based regional clusters shown
in **Figure 14**. Nevertheless, the generation of distinct regional marine radiocarbon calibration curves for the high latitude
Northeast Atlantic and the mid-latitude Northeast Atlantic (i.e. Iberian Margin) already emerges as a particularly promising
prospect.

### 5. Conclusions

K-Means, K-Medoids and hierarchical clustering reveal distinct regions of coherent R-age behaviour in the surface ocean,
subject to a range of perturbations, from seasonal to millennial timescales. In this context, the optimal value of K (the number
of clusters) is difficult to define robustly *a priori* and appears to depend on the method and the input data selected. The regional
clusters that are obtained, across the range of modelled oceanographic perturbations investigated, tend to cohere in a broadly
consistent manner with specific geographic domains, which in turn appear to reflect fundamental oceanographic and/or
seasonal controls on relevant processes such as sea ice variability, upwelling/mass divergence, etc. Clustering thus confirms
geographic controls on the variability in R-ages and their offset from the global mean surface ocean R-age (Stuiver et al.,
1986).  At larger spatial scales, clustering reveals broadly basin-scale associations in the 'character' (shape) of R-age
variability. These large-scale 'shape based' clusters may be further sub-divided into regional amplitude-based sub-clusters.
Comparisons within and between different model simulations, different time scales, and different models, indicate that
calibration curves constructed at appropriate locations, representative of the regional sub-cluster medoids/centroids, would
yield significantly more accurate calibrated radiocarbon dates than provided by the standard approach that assumes constant



delta-R values. Furthermore, a tentative application of these methods to existing R-age data identifies similar regional associations as compared to the numerical model outputs. Substantially more, and better resolved, R-age reconstructions, covering more of the worlds' ocean basins will be needed before robust regional radiocarbon calibrations can be fully tested

and applied. Nevertheless, based on our results, machine learning appears to be a promising approach to the problem of defining regional marine radiocarbon calibration curves. At present, the mid- and high-latitude sectors of the northeast Atlantic emerge as the most promising regions for initial progress in this regard.

## Code and Data availability

The scripts and complete python environment specifications used in this study are hosted at:

https://gitlab.com/earth15/ocean_data_clusters (see also Table S1 of the supplement). Data and model simulations used in this study have been previously published and are available via the referenced sources.

## Author Contributions

LCS designed the study. ACM developed the methods and code for data analysis in parallel with LCS. LM provided access to UVIC model outputs and assisted in their interpretation. LCS prepared the manuscript with contributions from all co-authors.

## Competing interests

The authors declare that they have no conflict of interest.

## Acknowledgments

This work was supported by NERC grant NE/V011464/1. The transient UVic simulations were performed at the Australian National Computational Infrastructure with support from the National Computational Merit Allocation Scheme. We are
grateful to Eric Galbraith for making the outputs from the CM2Mc model available for analysis.

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
