# Peer review of "Towards the construction of regional marine radiocarbon calibration curves: an unsupervised machine learning approach"

_Geochronology, 2023_

## Referee Comment (RC2)

Review of:
*Towards the construction of regional marine radiocarbon calibration curves: an unsupervised machine learning approach*

*Geochronology Discussions*
*Feb 2024*
*Referee: T J Heaton*

**Summary**

This is an interesting, and stimulating, paper which aims to use model simulations to identify ocean regions which appear to have similar levels of $^{14}$C depletion. The authors wish to make the case for regional marine radiocarbon calibration curves. They suggest that one might aim to identify suitable partitions for those regions using output from a computer model – inherently trying to cluster together locations for which the models generate similar output together.

I think this is an interesting and novel idea, although there are definitely challenges to overcome. Most directly, whether the current ocean circulation models are capable of providing reliable regional estimates and how/whether they account sufficiently for the degree of uncertainty regarding the behaviour of the oceans, fine-scaled circulation, the extent of sea-ice, and past changes to the carbon cycle.

In this paper the authors consider two ocean circulation models. One models a single year at high temporal precision and so I am not entirely sure if this is particularly relevant to a radiocarbon calibration curve providing at-best annual resolution estimates of oceanic depletion (but more likely somewhat coarser). The other model (UVIC) can be run over longer periods of time (they run it for about 15,000 years) and so would seem much more relevant to the problem at hand.

They run UVIC under two different scenarios with the aim of seeing if the regional clusterings of the set of simulated surface ocean depletion time series is consistent with different forcings. While running different carbon cycle scenarios certainly captures some potential variability, it is unclear whether other aspects of the model remain the same, and therefore to what extent there remains model specificity in the clusterings. I am not an expert in the details of the modelling, but I presume some of the parameters/interactions/fundamental underlying ocean circulation structures remains similar in both runs reliant even though they are inherently somewhat uncertain. As such one might need to be slightly cautious that other models might provide different clusterings.

Interestingly, both UVIC scenarios suggests similar clusters – effectively:
1) High-latitude (polar) Southern Ocean;
2) Pacific Ocean basin;
3) Atlantic Ocean basin;
4) High-latitude (polar) Arctic Ocean
These are the most immediate partitions one would select (indicating that the underlying UVIC model is doing sensible things).

However, there are perhaps some less expected features in that the regions in each cluster are not always geographically connected to one another (and in some cases even lie in different

hemispheres). I think this raises some practical questions as to whether such highly-disconnected locations should be grouped together into a single regional calibration curve (especially as the authors implicitly propose geophysical ideas as to why that general clustering, e.g. of the Southern Ocean, is appropriate in some of the discussion). Or whether we might need a combination of expert knowledge and ML to define regions.

I have no concerns with the application of the statistics. I think the paper will certainly provoke discussion and new ideas in the community. I certainly enjoyed reading it and it made me think. I would therefore recommend its publication.

I do have some comments, questions and suggestions that I hope will start discussion. I lay these out below. In general, my view is that regional marine calibration curves would be a fantastic and hugely important achievement, however we are still quite a way off being able to reliably generate them. This paper provides suggestions as to how we might navigate our way towards them.

**Main Points:**

**Existing/Related IntCal Recommendations on Regional Marine Calibration Curves:**

The IntCal group have aimed to explain to the community that using the current MarineXX curves for calibration requires the application of significant simplifications/approximations. In particular, users are generally required to estimate a value of $\Delta R_{20}(\theta)$, the offset between the localised surface-ocean depletion and the Marine20 curve, based on modern-day values and then consider that $\Delta R_{20}$ remains roughly constant over time. The idea here is that the main changes in oceanic $^{14}$C depletion occur at a global scale. This approximation of a constant $\Delta R_{20}(\theta)$ is discussed in detail in Heaton et al. (2023a) along with a broader discussion of the limitations of the Marine20 curve (which includes some of the issues the authors identify here).

Calibration of marine $^{14}$C samples is particularly challenging for polar (high-latitude) oceanic regions during glacial periods (as the authors highlight in Fig 1b) as modern day $\Delta R_{20}(\theta)$ are unlikely to be appropriate. During these times (and in these high-latitude locations) there may have been periods with substantial sea ice (that came and went) but which is not present in the modern day. This sea ice would have caused further localised $^{14}$C depletion and an increase in the MRA that is not represented in the global-scale Marine20 curve

To address this, the IntCal group have, in fact, already proposed a method to effectively perform regional marine calibration at high-latitudes. This can be found in Heaton et al (2023b). The proposed approach is a very simple (approximate) way of estimating upper and lower bounds on changes in $\Delta R_{20}(\theta)$ in glacial periods that uses the regional output of the LSG OGCM (Butzin et al. 2020).

At a given latitude, in Heaton *et al* (2023b) we propose first estimating a modern-day $\Delta R_{20}$. However, at high-latitudes during glacial periods, we recommend that this modern-day value may be too small and users will need to consider by how much it may be increased in a high-depletion $^{14}$C scenario. The amount by which we suggest it may need to be boosted is region (specifically latitude) dependent. We advise users with samples from glacial periods to calibrate under both a high (boosted- $\Delta R_{20}$ and low (modern day $\Delta R_{20}$) depletion scenario to provide bracketing calibrated ages which hopefully encapsulate the true calendar age. This latitudedependent adjustment effectively matches the partitioning proposed in this ML paper (where the clusters are effectively latitudinal bands concentrated on polar regions).

This proposed bracketing approach is not however mentioned in the submitted manuscript, and I think the representation of current calibration approaches might suggest to a reader that no suggestions to overcome the challenges of Marine calibration exist. The proposed Heaton et al. (2023b) approach is certainly simple, and coarse, but provides a first option until we get to a point where detailed and reliable regional marine calibration curves are possible. It may not allow highly-precise calibration (due to the high-level of uncertainty on past sea-ice extent) but hopefully will provide accurate calendar dating in polar regions.

**Notation:**

I would suggest that it would be extremely useful to add a subscript on all your estimated values of depletion to denote which calibration curve these are measured against, e.g., $MRA_{20}(\theta)$ and $\Delta R_{20}(\theta)$ if you are measuring against the IntCal20/Marine20 set of products, $MRA_{13}(\theta)$ if you are using the IntCal13/Marine13 products. See Heaton et al. (2023a) for our IntCal group advice/suggestion on this.

Without a subscript, it is unclear whether the plotted estimates of MRAs relate to the most recent set of IntCal20 curves or previous ones (as, e.g., Menviel *et al.* 2015 used for Figure 1b, would have originally been comparing with IntCal13 rather than IntCal20). Of course, the true marine depletion/offset is independent of the calibration curves, but the estimated values are not. The estimates will change with each update of calibration curve

This is particularly important when considering changes in $\Delta R(\theta)$ over time, as in the glacial period the Marine20 curve has changed significantly from Marine13. This will greatly affect the respective evolution of $\Delta R_{13/20}(\theta)$. The atmospheric curves have also changed somewhat which will affect the overall MRA but to a lesser extent.

Specifically, in Figure 1b, if these plots relate to changes in $\Delta R_{13}(\theta)$, then they could be quite different now. I am assuming that the main differences in the value of $\Delta R_{13}(\theta)$ at a specific location (i.e. the values plotted) will be during the late glacial. Here Marine13 assumed a constant global-scale MRA whereas Marine20 does not.

If Figure 1b is plotting $\Delta R_{13}(\theta)$ (i.e., using Marine13) then it may be that the maximal variation in Equatorial regions is now much smaller with $\Delta R_{20}(\theta)$ and Marine20. It would also be interesting to see if the maximal variation in $\Delta R_{20}(\theta)$ at higher latitudes with UVIC are similar to the suggested boosts/shifts proposed in Heaton et al. (2023b) using the LSG OGCM.

**General Philosophical Question:**

A philosophical question I might pose with the proposed approach is whether, until we are sure that the computer models accurately represent ocean circulation and past carbon cycle changes, can we use them to reliably cluster the ocean into regions where we can reliably group observational [14]C data to create regional marine calibration curves?

On the other hand, if we are confident that these models can represent circulation and carbon cycle, do we actually need data or can we just use the models themselves to produce calibration curves for any chosen location?

I would assume that to create the regional curves we would ideally collate records from multiple locations in any cluster. If so, do we need the prior (model-determined) clustering, or could we just create a lot of location specific curves from that data? Is a side/main benefit of the identified clustering is that it can to tell us about underlying properties of the computer models rather than to generate calibration curves?

**What to cluster on?**

A key question seems to be whether it is critical to generate regional marine calibration curves that need no $\Delta R$ adjustment for any location within that region (i.e., all locations have the same level of $^{14}C$/depletion)? Or is it more critical to generate calibration curves which might all need adjustment, but that adjustment is constant over time (i.e., they get the right evolution but perhaps offset by a constant amount).

I am little unclear which of these two options is being aimed for? If the concern is in the non-constancy of $\Delta R(\theta)$ over time, should the clustering be done on the variability rather the absolute value of the MRA. Is this the distinction between the normalised and un-normalised approaches to clustering. Are all the simulated outputs (for all locations) set to have mean zero for the normalised approach to clustering (i.e., a constant offset is considered identical) or is there also some renormalising of variance at every calendar age?

**Sea Ice**

My expectation is that a substantial factor in determining $^{14}C$ depletion in any marine location is sea ice. This would seem to be highly regional during glacial periods. Do we require detailed knowledge of sea ice extent and location for the computer models to reliably cluster locations together?

**Practicality of Disconnected Locations within a Cluster**

It seems potentially controversial to suggest one might create regional curves that cluster/group locations which aren't geographically close together (e.g., Fig 5 has clusters that contain high-latitude NH locations and high-latitude SH locations). Also, Fig 12 has some high-latitude NH regions which are clustered with Antarctic waters. It would be interesting to know whether this would be supported by the practical oceanographic community.

This is briefly discussed in Figure 14. Here, cluster 2 predominantly represents Antarctic waters but with a few regions in the high NH too. You implicitly seem to propose a potential link between the increase in MRA observed in this region to the Antarctic temperature. Are you suggesting this is perhaps the increased presence of sea ice? If it is sea ice in the SSoutehn Ocean then would affect the NH regions in the same cluster?

Is it appropriate to create a single regional curve for such geographically distinct regions? How might one strike a balance between a black-box ML clustering and incorporation of expert geoscientific knowledge?
* * *
**Technical Comments/Questions:**

**Clusters/Continuum** - How much of the variation is really on a continuum, and how much is it there really are distinct and separate clusters? The plots in Fig 5 suggest the clustering appears mainly based upon a scale of the mean overall MRA rather than hugely different shapes.

Interestingly, the clusters are predominantly latitude-based, is this basically indicating that the UVIC model has $\Delta R$ increasing more in polar regions than in more equatorial regions in glacial periods due to sea ice in those high-latitude locations? If so, this suggests UVIC and the LSG OGCM model concur with one another.

**Figure 14** – I do not entirely understand this plot. What is the difference between the red and blue lines for each identified cluster? You say they both represent the cluster – but isn't the point that there is supposed to be a single MRA for all sites in the cluster (not two). Also are the means (shown in solid lines) the UVIC model output or the averages of the observed data. I am presuming the latter, if so, are the observed MRAs in the specific clusters actually that similar – they seem to vary by 1000 $^{14}$C yrs between records within a cluster.

**Figure 13** – It is certainly the case that correct clustering /partitioning will provide you with more precise calibration curves. However, this seems a rather unfair comparison of Marine20 against the clustering approaches to make that point. In the central and RH panels, it seems you are effectively comparing simulations from UVIC with themselves; whereas in the LH panel you are comparing UVIC simulations with another entirely different model BICYCLE/Marine20. This is never likely to do as well. Furthermore, Marine20 aims to incorporate a much wider range of climate scenarios than the single climate scenario represented in the other panels by U-Tr.

**Specific (Minor) Comments:**

*Line 42* – I would say that perhaps the NH atmosphere is the only reservoir for which we have an entirely robust curve based upon direct observation (and even this is somewhat reliant upon the DCF in Hulu Cave being constant over time once we go back further than 14,000 cal yrs).

The SH calibration curve is, in large parts, reliant upon NH data and an assumption that the interhemispheric $^{14}$C gradient (IHG) has been roughly constant over time. Of course, the IHG is expected to be much less variable than the $^{14}$C depletion in the surface oceans, but we do still need more SH reference material to increase the precision of SH calibration.

Suggest one could reword the intro slightly to make clear that the SH calibration curve is certainly still a work in progress and more reference data is needed (and in fact, even the NH curve is reliant upon quite strong assumptions)

*Figure 1* – Panel a: Suggest you could be much clearer about precisely when this is a plot of in the title of the plot and the caption (also in the text you say it is pre-industrial, whereas in the caption you say it is modern and bomb-corrected? Which is it? Can you give a specific date as the overall MRA is highly variable from one year to the next. Also, it is unclear if this is modelled output or observation based – suggest could clarify what GLODAP is? Panel b needs clarification if this plots the changes in the estimated $\Delta R_{13}(\theta)$ or $\Delta R_{20}(\theta)$ (as explained in main comment).

*Line 261* – Do you mean cluster 1 forms a latitudinal band? Not longitudinal band

*Useage of the Term "Data"* – In general, I feel it would be useful to distinguish through the manuscript between genuine observed data and model output/simulations. Personally, I would restrict the use of the term *data* to refer to when one has actual observations. I would not describe output from a model as data – I think it is better to refer to it as modelled output, or a time series vector of simulated values. For example, I would not say you are clustering data (as that may suggest to readers that there are underlying observations) but rather you are clustering the vectorised model output.

***Figure 2*** – Suggest it could be made clearer this is an entirely artificial example to illustrate what clustering is. Perhaps this could be achieved just by creating a subsection explicitly called "A simple illustration of clustering" into which it could go. Initially I was a bit confused if these were the clustering of the actual vectorised simulated time series (with the principal components as the two plotted axes). Also, it would seem for Fig 2 as though 3 clusters is most appropriate to represent the data, rather than 4, so a bit unclear how it fits with the surrounding section about how you chose the optimal number of clusters.
* * *
**References:**

Heaton TJ *et al.* (2023a) 'A Response to Community Questions on the Marine20 Radiocarbon Age Calibration Curve: Marine Reservoir Ages and the Calibration of $^{14}$C Samples from the Oceans', *Radiocarbon*, 65(1), 247–273. doi:10.1017/RDC.2022.66.

Heaton TJ *et al.* (2023b) 'Marine Radiocarbon Calibration in Polar Regions: A Simple Approximate Approach using Marine20', *Radiocarbon*, 65(4), 848–875. doi:10.1017/RDC.2023.42.

Butzin, M., Heaton, T. J., Köhler, P., & Lohmann, G. (2020). A Short Note on Marine Reservoir Age Simulations Used in IntCal20. *Radiocarbon*, *62*(4), 865–871. https://doi.org/DOI: 10.1017/RDC.2020.9

---

## Author Comment (AC1)

**Response to review comments on gchron-2023-26:**

**We are very grateful for the comments provided by the reviewers and commentator, which we try to address below. We include the full commentaries received (underlining the parts of text that raise a question or appear to demand a response), along with our responses embedded inset in bold (red text).**

**Reviewer 1**

This is an interesting, and stimulating, paper which aims to use model simulations to identify ocean regions which appear to have similar levels of $^{14}$C depletion. The authors wish to make the case for regional marine radiocarbon calibration curves. They suggest that one might aim to identify suitable partitions for those regions using output from a computer model – inherently trying to cluster together locations for which the models generate similar output together.

I think this is an interesting and novel idea, although there are definitely challenges to overcome. Most directly, whether the current ocean circulation models are capable of providing reliable regional estimates and how/whether they account sufficiently for the degree of uncertainty regarding the behaviour of the oceans, fine-scaled circulation, the extent of sea-ice, and past changes to the carbon cycle.

In this paper the authors consider two ocean circulation models. One models a single year at high temporal precision and so I am not entirely sure if this is particularly relevant to a radiocarbon calibration curve providing at-best annual resolution estimates of oceanic depletion (but more likely somewhat coarser). The other model (UVIC) can be run over longer periods of time (they run it for about 15,000 years) and so would seem much more relevant to the problem at hand.

> **The reviewer's comment here (as well as those of the other commentators) suggest to us that we should clarify in a revised introduction that our method *intentionally* makes use of a diversity of different models, time-scales of variability, and drivers or variability. It is intended that millennial hosing runs, with different convection schemes in the Southern Ocean, are compared with the annual cycle in a glacial-like climate and an interglacial-like climate. This is because we wish to remain agnostic (in the models) as to what actually happened in the past, and instead assess only the coherence of spatial R-age variability subject to a wide range of different perturbation types/durations. The idea is to probe the limits of regional coherence subject to as wide a range of perturbations as possible. We should expand on the range of perturbations in future using a wider array of models and runs, but here we limit ourselves to a first assessment of the viability of the approach.**

They run UVIC under two different scenarios with the aim of seeing if the regional clusterings of the set of simulated surface ocean depletion time series is consistent with different forcings. While running different carbon cycle scenarios certainly captures some potential variability, it is unclear whether other aspects of the model remain the same, and therefore to what extent there remains model specificity in the clusterings. I am not an expert in the details of the modelling, but I presume some of the parameters/interactions/fundamental underlying ocean circulation structures remains similar in both runs reliant even though they are inherently somewhat uncertain. As such one might need to be slightly cautious that other models might provide different clusterings.

Interestingly, both UVIC scenarios suggests similar clusters – effectively:
1. 1) High-latitude (polar) Southern Ocean;
2. 2) Pacific Ocean basin;
3. 3) Atlantic Ocean basin;
4. 4) High-latitude (polar) Arctic Ocean

These are the most immediate partitions one would select (indicating that the underlying UVIC model is doing sensible things).

> **Indeed, and we emphasise in the manuscript that this is also true of the two other CM2Mc runs, as well as for clusters obtained in UVIC and applied to CM2Mc, for which completely different forcings are applied. The diversity of context here lends strength to the coherence of the clusters. This important for our approach, because (outside of some patchy observations) we don't actually know what happened to R-ages in the past all over the global ocean, and likely will not be able to simulate past global R-ages correctly for some time to come.**

However, there are perhaps some less expected features in that the regions in each cluster are not always geographically connected to one another (and in some cases even lie in different hemispheres). I think this raises some practical questions as to whether such highly- disconnected locations should be grouped together into a single regional calibration curve (especially as the authors implicitly propose geophysical ideas as to why that general clustering, e.g. of the Southern Ocean, is appropriate in some of the discussion). Or whether we might need a combination of expert knowledge and ML to define regions.

> **This is a very good point. As noted in the manuscript (e.g. line 429), some of these distal regions are actually connected through physical processes, such as the connection between the Southern Ocean and the Easter equatorial upwelling regions (these R-age connections have been discussed by e.g. de la Fuente et al. 2015 and Skinner et al., 2015). However, not all of them are (e.g. the polar regions).**

> **Ultimately, the answer the question posed may be somewhat philosophical: from a physical process perspective, some distal and disconnected regions should clearly not be grouped together as representing action by the same *processes*; however, from a statistical perspective it might still be reasonable to group them together as representing a similar *pattern* of change. For example, two distal regions may show no significant change in delta-R, in which case they could in principle make use of the same radiocarbon calibration curve (this premise is the same as that which would justify applying Marine20 to both the tropical Pacific and the tropical Indian Ocean at locations that have the same modern delta-R). That said, if one feels that physical understanding (expert knowledge) outweighs statistical similitude, then a cautious approach might be to not lump them together. We would add a comment on this to the revised manuscript.**

I have no concerns with the application of the statistics. I think the paper will certainly provoke discussion and new ideas in the community. I certainly enjoyed reading it and it made me think. I would therefore recommend its publication.

> **We are grateful for the reviewer's comments and encouraged that he found it stimulating.**

I do have some comments, questions and suggestions that I hope will start discussion. I lay these out below. In general, my view is that regional marine calibration curves would be a fantastic and hugely important achievement, however we are still quite a way off being able to reliably generate them. This paper provides suggestions as to how we might navigate our way towards them.

**We completely agree with this perspective, which we indeed try to emphasize in our conclusions.**

**Main Points:**

*Existing/Related IntCal Recommendations on Regional Marine Calibration Curves:*
The IntCal group have aimed to explain to the community that using the current MarineXX curves for calibration requires the application of significant simplifications/approximations. In particular, users are generally required to estimate a value of $\Delta R_!{}''(\theta)$, the offset between the localised surface-ocean depletion and the Marine20 curve, based on modern-day values and then consider that $\Delta R_!{}''$ remains roughly constant over time. The idea here is that the main changes in oceanic $^{14}$C depletion occur at a global scale. This approximation of a constant $\Delta R_!{}''(\theta)$ is discussed in detail in Heaton et al. (2023a) along with a broader discussion of the limitations of the Marine20 curve (which includes some of the issues the authors identify here).

Calibration of marine $^{14}$C samples is particularly challenging for polar (high-latitude) oceanic regions during glacial periods (as the authors highlight in Fig 1b) as modern day $\Delta R_!{}''(\theta)$ are unlikely to be appropriate. During these times (and in these high-latitude locations) there may have been periods with substantial sea ice (that came and went) but which is not present in the modern day. This sea ice would have caused further localised $^{14}$C depletion and an increase in the MRA that is not represented in the global-scale Marine20 curve To address this, the IntCal group have, in fact, already proposed a method to effectively perform regional marine calibration at high-latitudes. This can be found in Heaton et al (2023b). The proposed approach is a very simple (approximate) way of estimating upper and lower bounds on changes in $\Delta R_!{}''(\theta)$ in glacial periods that uses the regional output of the LSG OGCM (Butzin et al. 2020).

At a given latitude, in Heaton *et al* (2023b) we propose first estimating a modern-day $\Delta R_!{}''$. However, at high-latitudes during glacial periods, we recommend that this modern-day value may be too small and users will need to consider by how much it may be increased in a high- depletion $^{14}$C scenario. The amount by which we suggest it may need to be boosted is region (specifically latitude) dependent. We advise users with samples from glacial periods to calibrate under both a high (boosted- $\Delta R_!{}''$ and low (modern day $\Delta R_!{}''$) depletion scenario to provide bracketing calibrated ages which hopefully encapsulate the true calendar age. This latitude-dependent adjustment effectively matches the partitioning proposed in this ML paper (where the clusters are effectively latitudinal bands concentrated on polar regions). This proposed bracketing approach is not however mentioned in the submitted manuscript, and I think the representation of current calibration approaches might suggest to a reader that no suggestions to overcome the challenges of Marine calibration exist. The proposed Heaton et al. (2023b) approach is certainly simple, and coarse, but provides a first option until we get to a point where detailed and reliable regional marine calibration curves are possible. It may not allow highly-precise calibration (due to the high-level of uncertainty on past sea-ice extent) but hopefully will provide accurate calendar dating in polar regions.

**We agree and should have discussed the Heaton et al. (2023) study in our introduction. We would address this in a revised manuscript. We should emphasise perhaps that we are proposing a different (complementary) take on the problem, that ultimately would be based on data (as for Intcal20) and would obviate the need to presume knowledge of past ocean circulation, carbon cycling, and climate, as required when adopting forward model simulations as the basis for calibration.**

*Notation:*

I would suggest that it would be extremely useful to add a subscript on all your estimated values of depletion to denote which calibration curve these are measured against, e.g., $MRA_!"(\theta)$ and $\Delta R_!"(\theta)$ if you are measuring against the IntCal20/Marine20 set of products, $MRA_{\#\$}(\theta)$ if you are using the IntCal13/Marine13 products. See Heaton et al. (2023a) for our IntCal group advice/suggestion on this.

> **This is a good suggestion; however, this only applies to the data presented at the end of the paper. We will update any text that refers to these data. The model results are simply referred to the respective modelled atmosphere and are not in any way related to the Intcal products.**

Without a subscript, it is unclear whether the plotted estimates of MRAs relate to the most recent set of IntCal20 curves or previous ones (as, e.g., Menviel *et al.* 2015 used for Figure 1b, would have originally been comparing with IntCal13 rather than IntCal20). Of course, the true marine depletion/offset is independent of the calibration curves, but the estimated values are not. The estimates will change with each update of calibration curve
This is particularly important when considering changes in $\Delta R(\theta)$ over time, as in the glacial period the Marine20 curve has changed significantly from Marine13. This will greatly affect the respective evolution of $\Delta R_{\#\$/!"}(\theta)$. The atmospheric curves have also changed somewhat which will affect the overall MRA but to a lesser extent.
Specifically, in Figure 1b, if these plots relate to changes in $\Delta R_{\#\$}(\theta)$, then they could be quite different now. I am assuming that the main differences in the value of $\Delta R_{\#\$}(\theta)$ at a specific location (i.e. the values plotted) will be during the late glacial. Here Marine13 assumed a constant global-scale MRA whereas Marine20 does not.

> **Again, Figure 1b refers to model outputs and therefore does not refer to any Intcal products, but rather to the modelled atmospheric radiocarbon.**

If Figure 1b is plotting $\Delta R_{\#\$}(\theta)$ (i.e., using Marine13) then it may be that the maximal variation in Equatorial regions is now much smaller with $\Delta R_!"(\theta)$ and Marine20. It would also be interesting to see if the maximal variation in $\Delta R_!"(\theta)$ at higher latitudes with UVIC are similar to the suggested boosts/shifts proposed in Heaton et al. (2023b) using the LSG OGCM.

> **This is an interesting suggestion, but again it is crucial to note that we are not trying to simulate past R-ages: we are only interested in their spatial coherence subject to a variety of perturbations. This is why we compare annual cycles in different climate states with hosing experiments under different buoyancy forcings. Notably, Heaton et al. (2023b) did compare with one of the several R-age datasets included in Skinner et al. (2019), to show that observations confirm that the extreme scenarios of the LSG-OGCM do broadly bound 'reality', though with offsets as high as ~500 yrs.**

*General Philosophical Question:*
A philosophical question I might pose with the proposed approach is whether, until we are sure that the computer models accurately represent ocean circulation and past carbon cycle changes, can we use them to reliably cluster the ocean into regions where we can reliably group observational [14]C data to create regional marine calibration curves?

> **This is indeed an important question, but more so for those approaches that rely on forward modelling of the past (e.g. Alves et al., 2019; Heaton et al., 2023b). Crucially, our approach differs from such approaches in that we aim specifically to avoid needing to simulate past R-ages, and therefore also aim to**

**avoid requiring our models to accurately represent the past. Rather, what we require is that the models adequately represent the relevant *processes*, such that we may explore a sufficient range of perturbations to provide robust regional associations, regardless of the drivers and history of past R-age variability.**

**Given that the other commentaries received also appear to have misunderstood that we are trying to simulate past R-ages changes, we will incorporate a clearer discussion of this issue in both the introduction (to clarify our approach and aims), and in the conclusions (to clarify its limitations and extensions).**

On the other hand, if we are confident that these models can represent circulation and carbon cycle, do we actually need data or can we just use the models themselves to produce calibration curves for any chosen location?

**The premise of our proposed approach (and a firm view of at least one of us) is that forward simulations of *past* R-age variability (i.e. 'reconstructing history in silico') will always be somewhat wrong and will continue to be *substantially* wrong for some time to come (specifically as regards past carbon/radiocarbon cycling, ocean circulation, sea ice etc). Marine20 is an hypothesis, not an empirical product. It is also worth noting that when taking into account mean ocean radiocarbon data even box-models are still unable to close the global radiocarbon budget since the last glacial period (e.g. Skinner et al, 2023; Kohler et al., 2022). Furthermore, (to our knowledge) no high-resolution numerical model with completely free physics and biogeochemistry has yet been able to correctly simulate atmospheric $CO_2$ (after decades of attempts, and a great deal of process understanding), let alone simulate ocean interior- or surface ocean radiocarbon observations at the same time (e.g. Kohler et al. 2022). Therefore, yes, data will certainly be needed! We can amplify on this question (briefly) in a revised manuscript, as suggested above.**

I would assume that to create the regional curves we would ideally collate records from multiple locations in any cluster. If so, do we need the prior (model-determined) clustering, or could we just create a lot of location specific curves from that data?

**Yes, this is the entire purpose of our approach: to inform on the '*regionalism*' of R-age variability (using modelled physics/biogeochemistry variability) in order to both define the ideal locations for a collection of observations that would constitute a regional calibration curve, and to define the region to which that curve might apply.**

Is a side/main benefit of the identified clustering is that it can to tell us about underlying properties of the computer models rather than to generate calibration curves?

**Potentially yes, though to achieve this we would probably need the situation to be flipped with regard to the prevalence of modelled versus observed variables/variability: i.e. we would need to be in a data-rich context, which is currently absent for most of the ocean. However, as we suggest in the conclusions (and as was indicated in Skinner et al., 2019), the Northeast Atlantic emerges as a good candidate for further work already.**

*What to cluster on?*
A key question seems to be whether it is critical to generate regional marine calibration curves that need no $\Delta R$ adjustment for any location within that region (i.e., all locations have

the same level of $^{14}$C/depletion)? Or is it more critical to generate calibration curves which might all need adjustment, but that adjustment is constant over time (i.e., they get the right evolution but perhaps offset by a constant amount).

> **This is essentially the original question raised by Minze Stuiver in 1986 when he proposed the delta-R approach (Stuiver et al., Radiocarbon, 1986). However, as he pointed out at the time, if there have been significant carbon-cycle and ocean circulation changes, you can't know *a priori* where in the world these will have left delta-R values unaffected (though 'passive' stratified, atmosphere-equilibrated regions in the tropical gyres are obviously good candidates – hence the hypotheses embodied in Marine98 etc.). The situation where all potential regional calibration curves are merely offset from each other by a constant value is simply the same situation as using a single constant delta-R value relative to a global mean curve (e.g. Marine20 etc.). Our method aims to do something very different by identifying regions where delta-R values were *not* constant over time but at least *changed in the same way*. This is a pre-requisite for defining a regional calibration curve and its realm of applicability. Of course, the approach leaves the actual changes in delta-R (and regional calibrations) unresolved, as they would need to be constrained through observations (e.g. as per Skinner et al., 2019, or indeed any other studies that have sought to determine past surface R-age changes).**

I am little unclear which of these two options is being aimed for? If the concern is in the non-constancy of $\Delta R(\theta)$ over time, should the clustering be done on the variability rather the absolute value of the MRA. Is this the distinction between the normalised and un-normalised approaches to clustering. Are all the simulated outputs (for all locations) set to have mean zero for the normalised approach to clustering (i.e., a constant offset is considered identical) or is there also some renormalising of variance at every calendar age?

> **Yes, exactly. We first normalise variability to zero mean to assess the similarity of the variability in various regions (i.e. 'normalised' data, yielding 'shape-based clusters'). We then look into those groupings to produce sub-clusters on absolute values (i.e. 'raw data', with non-zero mean, yielding 'amplitude-based' sub-clusters). This approach yields amplitude-based sub-clusters within the shape-based regional clusters, for which more localised calibrations could be derived, or else constant delta-Rs might be applied relative to a regional calibration derived for the normalised 'parent' cluster. As we discuss in the manuscript and show in Figure 13, the use of regional calibration curves is more precise and accurate than the use of a global calibration with a constant delta-R value.**

*Sea Ice*

My expectation is that a substantial factor in determining $^{14}$C depletion in any marine location is sea ice. This would seem to be highly regional during glacial periods. Do we require detailed knowledge of sea ice extent and location for the computer models to reliably cluster locations together?

> **We agree that sea ice is surely crucial: this is exactly what was proposed in Skinner et al. (2019), based on a comparison of northern and southern R-age variability as compared to polar sea-ice reconstructions. Clearly the models that accurately simulate sea ice dynamics, and associated changes in convection and air-sea exchange, will give the most accurate regional clusters. However, it is important to underline again that here we are *not* talking about accurate simulations of past changes that *actually occurred*, but simply of the**

**processes that control the variability of sea ice etc. and their impacts on R-ages. Our approach proposes to avoid the need for detailed knowledge of past sea ice extent and the need for accurate simulations of past variability, relying instead on the regional 'boundedness' of such changes, due to robust physical/biogeochemical constraints (e.g. lines 374, 394, 470 of the original manuscript).**

*Practicality of Disconnected Locations within a Cluster*
It seems potentially controversial to suggest one might create regional curves that cluster/group locations which aren't geographically close together (e.g., Fig 5 has clusters that contain high- latitude NH locations and high-latitude SH locations). Also, Fig 12 has some high-latitude NH regions which are clustered with Antarctic waters. It would be interesting to know whether this would be supported by the practical oceanographic community.

**Please see our response above on this topic: we agree that from a physical process perspective such distal clustering may be implausible (outside of some distal regions that are indeed linked by physical processes, such as upwelling of sub-polar polar mode waters), while from a statistical perspective it may still be reasonable. Indeed, the use of a global average calibration curve such as Marine20 arguably presents the very same issue.**

This is briefly discussed in Figure 14. Here, cluster 2 predominantly represents Antarctic waters but with a few regions in the high NH too. You implicitly seem to propose a potential link between the increase in MRA observed in this region to the Antarctic temperature. Are you suggesting this is perhaps the increased presence of sea ice? If it is sea ice in the SSoutehn Ocean then would affect the NH regions in the same cluster?

**No, we are simply pointing out that in the modelled variability, those polar regions appear to vary with similar *shapes* (just as the tropical gyres from different basins may vary in the same way, yet with no physical connection). As we discuss in the manuscript, the connection is simply through similar patterns of variability (e.g. due to sea ice presence/absence at high latitudes, but not at low latitudes), but not necessarily due to shared physical forcings.**

Is it appropriate to create a single regional curve for such geographically distinct regions? How might one strike a balance between a black-box ML clustering and incorporation of expert geoscientific knowledge?

**This is a good question (please see our responses above on this topic). Although the clusters are simply a statement of similarity of variability and not of physical connectedness, one could argue that it is a moot point. However, we agree that the ML-based clusters should indeed be 'rationalised' with respect to their practical application, especially where regionally distal and disconnected regions are grouped together (e.g. by virtue of showing very muted variability, or variability that is no different from the global mean, etc.). We will add a note of this to the revised manuscript.**

**Technical Comments/Questions:**
**Clusters/Continuum** - How much of the variation is really on a continuum, and how much is it there really are distinct and separate clusters? The plots in Fig 5 suggest the clustering appears mainly based upon a scale of the mean overall MRA rather than hugely different shapes.

> **It depends, and this is the point of first grouping on the basis of 'shape', and then on the basis of amplitude. In many cases (e.g. the division between northern and southern hemisphere, or the North Atlantic and Southern Ocean) the clustering is very clearly based on distinct shapes, and not just scale/amplitude.**

Interestingly, the clusters are predominantly latitude-based, is this basically indicating that the UVIC model has $\Delta R$ increasing more in polar regions than in more equatorial regions in glacial periods due to sea ice in those high-latitude locations? If so, this suggests UVIC and the LSG OGCM model concur with one another.

> **We discuss this in the manuscript (e.g. line 330-334), where we note that amplitudes tend to increase with latitude (for obvious physical reasons, e.g. Bard, GCA 1998), but that distinct shapes also occur in different basins and hemispheres. The fact that LSG and UVIC and CM2Mc should all concur with one another (and with many other models) is precisely why our approach should work: it relies on fundamental constraints due to physical processes, and NOT the accuracy of simulating what actually happened in the past.**

**Figure 14** – I do not entirely understand this plot. What is the difference between the red and blue lines for each identified cluster? You say they both represent the cluster – but isn't the point that there is supposed to be a single MRA for all sites in the cluster (not two). Also are the means (shown in solid lines) the UVIC model output or the averages of the observed data. I am presuming the latter, if so, are the observed MRAs in the specific clusters actually that similar – they seem to vary by 1000 [14]C yrs between records within a cluster.

> **As noted in the caption, the red and blue lines represent the mean trends for each shape-based cluster performed on the data. Indeed, there is a great deal of scatter within these data-based clusters and we discuss in the manuscript that this scatter (as well as the paucity of existing data) precludes the generation of precise data-based clusters for now. Nevertheless, we propose that the results are tentatively encouraging, given that distinct shapes are indeed identified, which cohere with our understanding of the thermal bi-polar seesaw and its link to sea-ice variability etc. (as noted previously by Skinner et al., 2019).**
>
> **As noted in response to Reviewer 2 below, we will update this figure to also include a grouping of the data based on the model-based clusters (which is less affected by the uncertainty and paucity of data, and more affected by the accuracy of the modelled regionalism).**

**Figure 13** – It is certainly the case that correct clustering /partitioning will provide you with more precise calibration curves. However, this seems a rather unfair comparison of Marine20 against the clustering approaches to make that point. In the central and RH panels, it seems you are effectively comparing simulations from UVIC with themselves; whereas in the LH panel you are comparing UVIC simulations with another entirely different model BICYCLE/Marine20. This is never likely to do as well. Furthermore, Marine20 aims to incorporate a much wider range of climate scenarios than the single climate scenario represented in the other panels by U-Tr.

> **No, this is not correct, in all panels we are always comparing the model with itself. We either use the modelled global mean as the reference (e.g. to represent what a Marine20-like approach would yield, in the model), or a regional shape-based cluster centroid, or a sub-regional amplitude-based sub-**

**cluster centroid.  There is an increase in accuracy/precision in each case (as expected).**

**Specific (Minor) Comments:**
*Line 42* – I would say that perhaps the NH atmosphere is the only reservoir for which we have an entirely robust curve based upon direct observation (and even this is somewhat reliant upon the DCF in Hulu Cave being constant over time once we go back further than 14,000 cal yrs).

**This is perhaps a question of uncertainty tolerance, since ~50yrs is perhaps neither here or there when calibrating dates that are based on small foraminifer samples form ~25ka BP.  However, we would strengthen the distinction between the NH atmosphere and the global atmosphere in a revised text.**

The SH calibration curve is, in large parts, reliant upon NH data and an assumption that the interhemispheric $^{14}$C gradient (IHG) has been roughly constant over time. Of course, the IHG is expected to be much less variable than the $^{14}$C depletion in the surface oceans, but we do still need more SH reference material to increase the precision of SH calibration. Suggest one could reword the intro slightly to make clear that the SH calibration curve is certainly still a work in progress and more reference data is needed (and in fact, even the NH curve is reliant upon quite strong assumptions)

**Agreed.**

*Figure 1* – Panel a: Suggest you could be much clearer about precisely when this is a plot of in the title of the plot and the caption (also in the text you say it is pre-industrial, whereas in the caption you say it is modern and bomb-corrected? Which is it? Can you give a specific date as the overall MRA is highly variable from one year to the next. Also, it is unclear if this is modelled output or observation based – suggest could clarify what GLODAP is? Panel b needs clarification if this plots the changes in the estimated $\Delta R\#\$(\theta)$ or $\Delta R!"(\theta)$ (as explained in main comment).

**We will add these specifications.**

*Line 261* **–** Do you mean cluster 1 forms a latitudinal band? Not longitudinal band

**No, we do mean a band whose long dimension runs across longitude, around the Southern Ocean.**

*Useage of the Term "Data"* **–** In general, I feel it would be useful to distinguish through the manuscript between genuine observed data and model output/simulations. Personally, I would restrict the use of the term *data* to refer to when one has actual observations. I would not describe output from a model as data – I think it is better to refer to it as modelled output, or a time series vector of simulated values. For example, I would not say you are clustering data (as that may suggest to readers that there are underlying observations) but rather you are clustering the vectorised model output.

**Yes, we agree, this is a good point.**

*Figure 2* – Suggest it could be made clearer this is an entirely artificial example to illustrate what clustering is. Perhaps this could be achieved just by creating a subsection explicitly called "A simple illustration of clustering" into which it could go. Initially I was a bit confused if these were the clustering of the actual vectorised simulated time series (with the principal components as the two plotted axes). Also, it would seem for Fig 2 as though 3 clusters is

most appropriate to represent the data, rather than 4, so a bit unclear how it fits with the surrounding section about how you chose the optimal number of clusters.

> **Agreed. We would probably opt to remove this illustrative figure from a revised manuscript in the interest of saving space, as it brings relatively little.**

**References:**

Heaton TJ *et al.* (2023a) 'A Response to Community Questions on the Marine20 Radiocarbon Age Calibration Curve: Marine Reservoir Ages and the Calibration of $^{14}$C Samples from the Oceans', *Radiocarbon*, 65(1), 247–273. doi:10.1017/RDC.2022.66.
Heaton TJ *et al.* (2023b) 'Marine Radiocarbon Calibration in Polar Regions: A Simple Approximate Approach using Marine20', *Radiocarbon*, 65(4), 848–875. doi:10.1017/RDC.2023.42.
Butzin, M., Heaton, T. J., Köhler, P., & Lohmann, G. (2020). A Short Note on Marine Reservoir Age Simulations Used in IntCal20. *Radiocarbon*, *62*(4), 865–871. https://doi.org/DOI: 10.1017/RDC.2020.9

**Reviewer 2**
Marza et al., present an analysis in which they use clustering algorithms to investigate the spatial and temporal variability of marine radiocarbon reservoir ages in model simulations. The authors demonstrate that this type of analysis has a strong potential for improving marine radiocarbon calibration curves compared to the current standard approach of applying constant reservoir age corrections to the global marine calibration curve. I find the analysis interesting and agree there is a strong potential for this approach to lead to better radiocarbon-based age models for marine cores.

> **We are very grateful for the comments and corrections provided by the reviewer, which we try to address below.**

**Comments:**
Overall, my comments are minor. The problem and questions are well-introduced and the results are explained mostly coherently. My one complaint would be that I hoped the authors could go further in demonstrating how the cluster results can already inform marine core age models.
If this analysis were conducted on a transient simulation of the deglaciation, then the sediment-based R-age estimates shown in Fig. 14 could be used as validation of the k-medoids results, and the regional R-age curves from the cluster analysis would probably provide already provide useful estimates of reservoir ages for paleoceanographers. However, the interval selected from MIS 3 makes validation with proxy data difficult, and the regional R-age curves are probably less useful given large uncertainties in 14C ages from MIS3. Clearly, redoing the analysis on a new set of simulations is outside the scope of the current study, however it would be worth including some justification in the methods about why the two model datasets were selected and not a transient simulation of the deglaciation.

> **We are grateful for the suggestion. As noted above in response to Reviewer 1 (and to the comment from Katy Sparrow below), directly simulating deglacial R-age evolution is indeed one way to address the marine 14C calibration conundrum. This approach has its own major challenges. However, we should emphasize that we actually aim specifically *not* to do this. Rather, our approach is to use simulations of variously perturbed ocean states to simply inform on the spatial coherence (or otherwise) of R-age changes subject to those perturbations, and thus to assess the viability of defining 'regions' of coherent behaviour for which regional calibration curves could be produced eventually. Therefore, it is important for our approach to use a diversity of simulations that**

**do not necessarily capture the history of deglaciation. We would aim to make this much clearer in the introduction to a revised manuscript.**

The brief section on the sediment-derived R-ages feels somewhat disconnected from the results of the models. I was hoping to see some validation that could show that the clusters found in the models are somehow reasonable when real datasets are considered. To that end – what does Fig. 14 look like if you apply the clusters from Fig 10 rather than calculating new clusters? Are the results similar to what you obtained from clustering the proxy timeseries? This would provide a good test of the cluster results and show their utility for interpreting paleo records of reservoir ages.

**Indeed, showing that the clusters found in the models are somehow reasonable when real datasets are considered is exactly what we wished to do at the end of the manuscript. The reviewer makes an excellent suggestion. We had opted to attempt to cluster the existing data and compare it to the clusters from the models. However, in our revised manuscript we will also apply the clusters suggested by the models and assess the degree of coherence of the R-age data thus grouped. The former is a more difficult test to pass, given the paucity and quality of the existing data, but it is indeed interesting to compare both approaches.**

The authors suggest that applying these cluster methods to data-based reservoir ages could help develop regional reservoir age curves, and I agree. However, I am also wondering if it is possible to already gain some improvements by applying the results of the cluster analysis shown here without the need for generating new proxy R-age records. Please comment, are the k-medoids timeseries of R-ages already useful for constructing regional calibration curves, why or why not? Are there next steps that can be suggested to achieve regional calibration curves based on modelling/statistical methods?

**We agree that it would be nice to already move toward regional calibrations using the existing data, and Skinner et al. (2019) did propose an empirical regional calibration for the Iberian Margin. However, the problem that we address here is the definition of what counts as a 'region', including what would count as the 'Iberian Margin region' for the purposes of radiocarbon calibration. As we state in the manuscript, the ability to generate calibration curves for the regions suggested by the clustering is currently hampered by data availability (and uncertainties); however, we do indicate that the Northeast Atlantic stands out as a region where immediate progress could be made. In a revised manuscript we can amplify on this with more specific suggestions, though we should leave the generation of such a calibration curve and the definition of its region of applicability to a future study.**

Another idea that comes to mind for the application of these results could be a tool in which researchers can input a coordinate of a sampling location and then generate a map of the expected similarity of R-ages. With this information, one could select sites that have useful information about R-ages for your site of interest. Maybe an idea for future work…

**This is an excellent suggestion, and one that we (or others) could pursue in future. For now, we feel that our results provide strong support for the approach, but do not yet provide a robust enough definition of expected regional associations. We do plan to do this (i.e. propose definite calibration 'regions') in a forthcoming study that would make use of a much larger array of modelling and data-informed regional clusters.**

**Minor Comments**

Fig 1 - Define U-Tr in caption. Over what timespan is this maximum variability calculated?

**We will add this to a corrected manuscript (the model runs and durations are set out in Table 1, but we can re-iterate them to avoid requiring the reader to look them up).**

Line 240 – Line 245: I believe there are several discrepancies between how the text describes the figure, and what is shown in the figure and written in the caption. This makes it very difficult to follow. Some examples are below, but please check carefully.
Line 240 - Elbow is shown at K=6 in the figure, but written as K = 5 in the text.
Line 243 – K-medoids are on the right side of the plot, change "(Figure 4 left hand plots, dashed lines)" to say right hand plots
Line 244 - "when using normalized data (Figure 4)" – should it say "unnormalized data"?

**We are grateful that these corrections and/or ambiguities have been spotted; we will correct or clarify all of the above in a revised manuscript.**

Line 245 – I am unable to parse the meaning of the concluding sentence of this paragraph.

**We will clarify or remove the sentence: all we mean to say is that the optimal value of K is as subjective/woolly for the amplitude-based sub-clusters as it is for the shape-based clusters.**

Fig. 5 - Label time step units. Label A, B, C as noted in text.

**Thank you, yes we will correct these errors.**

Line 279 – Suggest deleting relatively.

**Agreed (it is not 'insensitive' to amplitudes, but the meaning is probably clear enough).**

Fig. 9&10 – label time step units

**Yes, we will correct this.**

Fig. 11 – How was 7 chosen as the number of clusters? Based on the dendrogram, 4 clusters appears more logical to me (greater gap in the distances there), and would allow a more direct comparison with Fig 9.

**These sections were for illustrative purposes, but we agree that it would be better to illustrate 4 clusters, as is used ultimately in most of our analyses; we will correct this.**

Personally, I find Fig. S3 more interesting (comparing results from two different models and two different timescales of variability) than some of the plots included in the main text (eg Fig 12, Fig 6).

**We agree, but thought we might be pushing our luck with the number of figures!  We will include the other intra- and inter model comparisons in the revised main text, combined into a single Figure 12 with three panels.**

It would be interesting to also compare the K-medoids timeseries obtained from the U-Tr raw data with those shown in Fig. 9 and 10. I.e. what happens if you just use un-normalized data

rather than doing a two step procedure with normalized data and then subclusters on raw data?

> **This is essentially what we try to show in Figure 13, where the mean R-age offset and the variance in the offsets are shown for each location in the surface ocean when the local R-age is compared to: 1) the global mean (i.e. as per the delta-R approach); 2) the raw cluster medoid/centroid; and 3) the sub-cluster medoid/centroid.**

Line 417 – I think Figure 1b should rather be 1a

> **No, it should be 1b (though we will label the panels in a corrected Figure 1): we are referring to how 'wrong' the assumption of the modern delta-R would be, in the model simulation (across all time, on average).**

**COMMENT BY KATY SPARROW**

This article by Marza et al. deals with an important topic for studies involving the radiocarbon dating of marine samples. It is widely recognized within the scientific community that global curves are not ideal for the calibration of ages obtained from marine material. However, the authors appear to have overlooked the work of Alves et al. (2019), which directly addresses the same issue of marine calibration in radiocarbon dating and discusses the limitations and challenges associated with constructing regional marine calibration curves.

Alves, E.Q., Macario, K.D., Urrutia, F.P., Cardoso, R.P. and Ramsey, C.B., 2019. Accounting for the marine reservoir effect in radiocarbon calibration. *Quaternary Science Reviews*, 209, pp.129-138, https://doi.org/10.1016/j.quascirev.2019.02.013

> **We are grateful for the comment, and for reminding us of this study, which we should have cited. It indeed serves as an example of a very different approach to the marine radiocarbon calibration problem, which seeks instead to simulate past R-ages over time and space, based on a priori knowledge (or perfect/adequate simulations) of past ocean circulation, carbon cycling, radiocarbon production etc. We propose a very different approach, to which the study of Alves et al (2019) indeed serves as an excellent counterpoint, which we were remiss to not mention.**